



# Ionic aluminium concentrations exceed thresholds for aquatic health in Nova Scotian rivers

Shannon Sterling[1], Sarah MacLeod[2], Lobke Rotteveel[1], Kristin Hart[1], Thomas A. Clair[1], Edmund A. Halfyard[3]

[1] Sterling Hydrology Research Group, Department of Earth Sciences, Dalhousie University, Halifax, Nova Scotia, Canada

[2] Coastal Action, Lunenburg, Nova Scotia, Canada

[3] Nova Scotia Salmon Association, Chester, Nova Scotia, Canada

*Correspondence to*: Shannon Sterling (shannon.sterling@dal.ca)



**Abstract.** Cationic aluminium species are toxic to terrestrial and aquatic life. Despite decades of acid

emission reductions, accumulating evidence shows that freshwater acidification recovery is delayed in

locations such as Nova Scotia, Canada. Further, spatial and temporal patterns of labile cationic forms

of aluminium ($Al_i$) remain poorly understood. Here we increase our understanding of $Al_i$ spatial and

temporal patterns by measuring $Al_i$ concentrations in ten streams in acid-sensitive areas of Nova

Scotia over a four-year time period. We observe widespread and frequent occurrences of $Al_i$

concentrations that exceed toxic thresholds ($>15\ \mu g\ L^{-1}$). $Al_i$ patterns appear to be driven by known

$Al_i$ drivers - pH, dissolved organic carbon, dissolved aluminium, and calcium - but the dominant

driver and temporal patterns vary by catchment. Our results demonstrate that elevated $Al_i$ remains a

threat to aquatic ecosystems. For example, our observed $Al_i$ concentrations are potentially harmful to

the biologically, economically, and culturally significant Atlantic salmon (*Salmo salar*).

## 34  1 Introduction

Freshwater acidification caused elevated concentrations of cationic aluminium ($Al_i$) at the end

of the last century that led to increased freshwater and marine mortality and, ultimately, the extirpation

of native Atlantic salmon (*Salmo salar*) populations in many rivers (Rosseland et al., 1990), for example

in Scandinavia (Henriksen et al., 1984, Hesthagen and Hansen, 1991), the eastern USA (Monette and

McCormick, 2008, Parrish et al., 1998), and Nova Scotia, Canada (Watt, 1987). Following reductions

in anthropogenic sulfur emissions in North America and Europe since the 1990s, many rivers showed

steady improvements in annual average stream chemistry (Evans et al., 2001, Monteith et al., 2014,



Skjelkvåle et al., 2005, Stoddard et al., 1999, Warby et al., 2005), including reduced concentrations of

$Al_i$ in the USA (Baldigo and Lawrence, 2000, Buchanan et al., 2017, Burns et al., 2006) and Europe

(Beneš et al., 2017, Davies et al., 2005, Monteith et al., 2014). However, recent evidence highlights

delayed recovery from acidification in some areas (Houle et al., 2006, Warby et al., 2009, Watmough

et al., 2016), including SWNS (Clair et al., 2011), raising concerns about elevated $Al_i$ concentrations.

Aluminium (Al) toxicity can be caused by both precipitated and dissolved forms in

circumneutral waters (Gensemer et al., 2018); however, the cationic species of Al, such as $Al^{3+}$,

$Al(OH)_2^{1+}$, and $Al(OH)^{2+}$ are considered to be the most labile and toxic to salmonids as they bind to the

negatively charged fish gills causing morbidity and mortality through suffocation (Exley et al., 1991),

reducing nutrient intake at gill sites, and altering blood plasma levels (Nilsen et al., 2010). Further, the

effects of sub-lethal exposure to freshwater Al elicits osmoregulatory impairment (Monette and

McCormick, 2008, Regish et al., 2018) which reduces survival in the hypertonic marine environment

(McCormick et al., 2009, Staurnes et al., 1996).   Elevated concentrations of $Al_i$ are also toxic to other

freshwater and terrestrial organisms (Boudot et al., 1994, Wauer and Teien, 2010), such as frogs and

aquatic birds (Lacoul et al., 2011).

Al speciation varies with pH (Helliweli et al., 1983, Lydersen, 1990), where positive Al species

dominate over neutral and negative species below pH 6.3 at 2 °C and below pH 5.7 at 25 °C (Lydersen,

1990), with the most toxic Al species, $Al(OH)_2^{+1}$ (Helliweli et al., 1983) dominating Al speciation

between pH 5.0–6.0 at 25 °C, and 5.5–6.5 at 2 °C (Lydersen, 1990). Thus, the toxicity of Al increases

with increased pH up to the formation of gibbsite (Schofield and Trojnar, 1980). Additionally, colder

waters will have a higher proportion of toxic species at higher pH values than warmer waters (Driscoll

and Schecher, 1990). The bioavailability of Al is reduced by the presence of calcium (Ca) (Brown,

1983), which can occupy the negatively charged gill sites, and dissolved organic carbon (DOC), which





occludes $Al_i$ through the formation of organo-Al complexes ($Al_o$) that are nontoxic to fish (Erlandsson

et al., 2010).

Despite being the most common metal on Earth's crust, Al is usually immobilized in clays or

hydroxide minerals in soils. Rates of Al release into soil water from soil minerals increase with three

drivers: 1) low soil pH, 2) low soil base saturation, and 3) high soil DOC concentrations. Lowered pH

increases Al solubility and observations confirm that $Al_i$ concentrations are negatively correlated with

pH (Campbell et al., 1992, Kopáček et al., 2006). Low levels of base saturation can cause charge

imbalances resulting in the release of Al into soil waters from clay particles, and later into drainage

waters (Fernandez et al., 2003) and chronic acidification thus shifts available exchangeable cations in

the soil from Ca and magnesium (Mg) towards Al (Schlesinger and Bernhardt, 2013, Walker et al.,

1990). Higher concentrations of DOC in soil water increase the release of Al through two mechanisms:

1) as an organic acid, DOC decreases soil pH, thus increasing Al release (Lawrence et al., 2013), and

2) by forming organic complexes with $Al_i$ it maintains a negative Al concentration gradient from the

cation exchange sites to the soil water, increasing rates of Al release (Edzwald and Van Benschoten,

1990, Jansen et al., 2003). Field studies confirm Al concentrations to be positively correlated with DOC

(Campbell et al., 1992, Kopáček et al., 2006) although at higher concentrations of DOC, Al may be

organic-complexed and less toxic to aquatic organisms (Witters et al., 1990).

Once mobilized in soil waters, export of $Al_i$ to drainage waters requires anions to maintain

charge balance. Storm events have been shown to increase $Al_i$ export due to added anions (e.g., $Cl^-$,

$SO_4^{2-}$, $F^-$), and from the movement of flow paths to shallower soil horizons where more Al may be

available for transport.  For example, from 1983 to 1984, Al concentrations for the River Severn in

Wales increased ten-fold during the stormflow peak compared to the baseflow (Neal et al., 1986).





However, the association of increased $Al_i$ concentrations with storm flow is not consistent in the

literature (DeWalle et al., 1995, McKnight and Bencala, 1988).

Annual patterns of $Al_i$ typically show a peak, but the timing of the peak varies. In some areas,

$Al_i$ concentrations peak in the spring and winter, correlated with flow peaks, such as in Quebec

(Campbell et al., 1992), Russia (Rodushkin et al., 1995), and along the Czech-German border (Kopacek

et al., 2000, Kopáček et al., 2006). In other areas, Al concentrations were found to be higher in the

summer such as in Virginia, USA (Cozzarelli et al., 1987). If the timing of peak $Al_i$ concentrations

coincides with sensitive stages of aquatic organisms, the potential for large biological impacts is high.

Our understanding of spatial and temporal trends of $Al_i$ is limited by the relative paucity of

samples: $Al_i$ is not measured as part of standard analyses. Our understanding is also limited by the

difficulty in comparing the wide variety of methods for estimating $Al_i$; different definitions, often

operational, of toxic Al include inorganic Al, inorganic monomeric Al, labile Al, $Al^{3+}$, and cationic Al

(Table A1). Definitions for both inorganic monomeric Al and cationic Al include all positively charged

species of Al.

Acid sensitive areas of NS, here abbreviated as $NS_A$ (see Clair et al., 2007), with once-famous

wild Atlantic salmon populations, were heavily impacted by acid deposition at the end of the last

century, which originated from coal burning in central Canada and Northeastern USA (Hindar, 2001,

Summers and Whelpdale, 1976).  $NS_A$ catchments are particularly sensitive to acid deposition due to

base cation-poor and slowly weathering bedrock that generates thin soils with low acid neutralizing

capacity (ANC), extensive wetlands, and episodic sea salt inputs (Clair et al., 2011, Freedman and Clair,

1987, Watt et al., 2000, Whitfield et al., 2006). A 2006 fall survey found that $Al_i$ concentrations in NS

exceeded the 15 μg $L^{-1}$ toxic threshold suggested by the European Inland Fisheries Advisory Council

(EIFAC) for aquatic health in seven of 42 rivers surveyed (Dennis and Clair, 2012). However, apart





from this study, little is known about the regional extent and patterns of $Al_i$. Here, we aim to increase

our understanding of current $Al_i$ spatial and temporal patterns in relation to toxic thresholds, and to

identify potential drivers by conducting a four-year survey of $Al_i$ concentrations in ten streams across

acid-sensitive areas of NS, Canada.

## 114   2  Materials and methods

### 115   2.1 Study area

We surveyed $Al_i$ concentrations at ten study catchments in $NS_A$, ranging from headwater to

higher-order systems: Mersey River (MR), Moose Pit Brook (MPB), Pine Marten Brook (PMB), Maria

Brook (MB), Brandon Lake Brook (BLB), above the West River lime doser (ALD), Upper Killag River

(UKR), Little River (LR), Keef Brook (KB), and Colwell Creek (CC) (Table 1, Fig. 1 and 2). Our study

catchments are predominantly forested, draining slow-weathering, base-cation poor bedrock, producing

soils with low ANC (Langan and Wilson, 1992, Tipping, 1989). The catchments also have relatively

high DOC concentrations (Ginn et al., 2007) associated with the abundant wetlands in the region (Clair

et al., 2008, Gorham et al., 1986, Kerekes et al., 1986).

### 124   2.2 Data collection and analysis

We measured $Al_i$ concentrations at three of the ten catchments from April 2015 to September

2017 (MR, MPB, PMB), on a weekly to monthly frequency during the snow free season (approximately

April to November, Table A2). In 2016-2018, seven sites were added and sampled every two weeks to

monthly during the snow-free season.





$Al_i$ sampling events comprise grab samples for lab analysis and in situ measurements of pH and

water temperature ($T_w$). We calculate $Al_i$ as the difference between dissolved Al ($Al_d$) and $Al_o$ following

Dennis and Clair (2012) and Poléo (1995) (Eq. 1), separating the species in the field to reduce errors

caused by changes in temperature and pH in transport from field to lab.

$$Al_i = Al_d - Al_o$$    (1)

$Al_d$ is measured as the Al concentration of a filtered sample and $Al_o$ is measured as the eluate

from passing filtered water through a 3 cm negatively charged cation exchange column (Bond Elut Jr.

Strong Cation Exchange Column). Samples were passed through the cation exchange column at a rate

of approximately 30 to 60 drops per minute. From this method, $Al_o$ is operationally defined as the non-

labile, organically-complexed metals and colloids, and $Al_i$ is defined as the positive ionic species of Al

(e.g., $Al^{3+}$, $Al(OH)^{2+}$, and $Al(OH)_2^+$).

Stream chemistry samples (50 ml) were collected using sterilized polyethylene syringes into

sterilized polyethylene bottles.  Samples for sulfate ($SO_4^{2-}$) analysis were not filtered. Trace metal

samples were filtered (0.45 µm) and preserved with nitric acid ($HNO_3$). Samples for DOC analysis were

filtered (0.45 µm) and transported in amber glass bottles containing sulfuric acid preservative ($H_2SO_4$)

to prevent denaturation. All samples were cooled to 7 °C during transport to the laboratories. Samples

were delivered to the laboratories within 48 hours of collection, where they were further cooled to ≤

4°C prior to analysis (Appendix D).

We examined correlations between $Al_i$ and water chemistry parameters: $Al_d$, Ca, DOC, pH,

$SO_4^{2-}$, $T_w$, fluoride ($F^-$), nitrate ($NO_3^-$), and runoff (where data are available). Correlations were analysed

within and across sites. For the purposes of this study, we use the toxic threshold of $Al_i$ at 15 ug L$^{-1}$, as

the majority of our pH observations were greater than or equal to 5.0 (Table A2, Appendix D3).





## 3 Results and discussion

### 3.1 Patterns of $Al_i$

$Al_i$ concentrations exceed toxic levels (15 ug $L^{-1}$) at all sites during the study period (Table A2).

Sites in the eastern part of the study area have the highest proportion of samples exceeding threshold

levels, including one site with 100% of samples in exceedance (Fig. 1). Mean $Al_i$ concentrations across

all sites range from 13–60 ug $L^{-1}$ (Table 1), with the highest mean concentrations also occurring in the

eastern part of the study area (Fig. 2). $Al_i$ concentrations exceed 100 ug $L^{-1}$ (approximately seven times

the threshold) at three sites (Table A2). In the sites with the longest and most frequent data collection

(MR and MPB), $Al_i$ concentrations exceed the toxic threshold in consecutive samples for months at a

time, particularly in the late summer (Fig. B1). Our $Al_i$ concentrations are consistent with the 6.9–230

161 ug $L^{-1}$ range of $Al_i$ concentrations measured across NS by Dennis and Clair (2012) and are higher than

162 concentrations measured in Norway from 1987–2010 (5–30 ug $L^{-1}$) (Hesthagen et al., 2016).

The percent of Al not complexed by DOC (% $Al_i/Al_d$) ranges from a minimum of 0.6% to a

maximum of 50%, with a median value of 10.7%, across all sites. These findings are similar to those

found NS by Dennis and Clair (2012) of the proportion of $Al_i$ in total aluminum ($Al_t$) (min. = 4%, max.

= 70.1%, med. = 12.4%), and less than those found by Lacroix (1989) (over 90 % $Al_o/Al_d$). $T_w$ and pH

have a significant positive correlation with $Al_i/Al_d$ (Table A3), consistent with an earlier observation

that Al toxicity increases with pH (Schofield and Trojnar, 1980). However, even when the percentage

of $Al_i/Al_d$ is low, $Al_i$ concentrations remain well above thresholds for toxicity (Fig. B4-B13). Previous

studies show $Al_i/Al_d$ is low during baseflow (Bailey et al., 1995, Murdoch and Stoddard, 1992,

Schofield et al., 1985), similar to our findings (Figs. B4-B13); more consistent year-round sampling is

needed to obtain a better picture of seasonal patterns in Al speciation in $NS_A$.



## 3.2 Potential $Al_i$ drivers

$Al_d$ is significantly ($\alpha = 0.05$) and positively correlated with $Al_i$ in seven of the ten study sites (ALD, KB, LR, MB, MPB, MR, PMB) (Fig. 3, Table A4), despite the high concentrations of DOC. $Al_i$ is also significantly and positively correlated with DOC in four sites (ALD, KB, MPB, MR) (Fig. 3, Table A4), consistent with other studies (Campbell et al., 1992, Kopáček et al., 2006). The positive correlation between DOC and $Al_i$ concentrations may suggest that the ability of DOC to mobilize $Al_d$ in soils is stronger than its ability to occlude $Al_i$ in streamwaters.

Ca is significantly and positively correlated with $Al_i$ at two sites (MPB, MR) (Fig. 3, Table A4). The positive relationship between Ca and $Al_i$ is the opposite of expectations. We hypothesize that this is due to the two study sites having very low Ca concentrations (mean concentrations below 1 mg $L^{-1}$), below which soil water Ca concentrations are too low to retard Al release. $T_w$ is also significantly positively correlated with $Al_i$ at two sites (MR, MPB) (Fig. 3, Table A4), likely reflective of the temperature-related drivers of Al concentration and speciation. Runoff is significantly and negatively correlated with $Al_i$ at one site MPB (Fig. 3, Table A4). Runoff data are available for only two of the study sites (MR, MPB) and so more runoff data are needed to improve our understanding of the relation between runoff and $Al_i$ in $NS_A$.

We did not observe the negative association between pH and $Al_i$ observed in previous studies (Campbell et al., 1992, Kopáček et al., 2006). pH is negatively correlated with $Al_i$ in four out of ten sites, but none of these relationships are statistically significant (Fig. 3, Table A4). We did observe a statistically significant positive relationship between pH and $Al_i/Al_d$; thus it seems that pH may play a more important role in determining the proportion of different Al species rather than the absolute value of $Al_i$ present in streamwaters.





F$^-$ has also been found to be a complexing agent that affects the speciation of Al at low pH levels

and relatively high concentrations of F$^-$ (>1 mg L$^{-1}$) (Berger et al., 2015). The concentrations of F$^-$ at

the study sites are mostly below this threshold (mean across all sites = 0.045 mg L$^{-1}$); however, there is

still a significant positive effect of F$^-$ on Al$_i$ concentrations across at two sites (KB, MPB) (Fig. 3, Table

A4). NO$_3^-$ and SO$_4^{2-}$ are also potential complexing ligands of Al; however, we did not observe any

correlation between Al$_i$ and either of these parameters, except for a significant negative correlation

between SO$_4^{2-}$ and Al$_i$ at MB.

The highest concentrations of Al$_i$ observed (> 100 ug L$^{-1}$) often occurred in early summer (late

June or early July in 2016-2018) when Al$_d$, Ca, and DOC concentrations had not yet reached their

annual peak (Table A2). The spring/summer extreme events occurred among the first exceptionally

warm days (> 21 °C) of the year, in dry conditions, and when the proportion of Al$_o$/Al$_d$ was low

(lowering to approximately 60-70% from higher levels of around 80-90%) (Figs. B4-B13). pH was not

abnormally low during these events (ranging from 4.8 to 6.13), Ca concentrations were low (less than

or equal to 800 µg L$^{-1}$) and DOC concentrations ranged from 15–21 mg L$^{-1}$. The observed peak in Al$_i$

concentrations during times of lower discharge contrasts with studies that found higher Al$_i$

concentrations during higher flow (Campbell et al., 1992, Kopacek et al., 2000, Neal et al., 1986,

Rodushkin et al., 1995). Further research is required to test hypotheses on why high Al$_i$ coincides with

high DOC and low flow periods.

**3.3 Possible seasonal groupings of Al$_i$ in NS$_A$**

In the two sites with the most samples, MPB and MR, groupings of data are visible that are

temporally contiguous, potentially indicating seasonally-dependent Al$_i$ behavior (Fig. 4). This is

supported by stronger linear correlations (r$^2$) among variables when grouped by "season" (Table 2); for





example, for the correlation between pH and $Al_i$ at MR, $r^2$ improves from 0.02 for year-round data (Fig.

B17) to up to 0.78 in season 1 (Fig. 4). The transition dates between the seasons are similar for the two

catchments, but not the same (Table A2), and vary by year. Here we propose initial characterization of

the potential "seasons"; more research is needed to test these hypotheses on seasonal divisions and their

drivers using larger datasets and Generalized Linear Mixed Model analysis to test for statistical

significance among the potential seasonal groupings.

Season 1 (approximately April/May) is coincident with snow-melt runoff and is characterized

by relatively low concentrations of $Al_i$ (2-46 ug L$^{-1}$), low pH (4.5-5.3), and lower concentrations of

most constituents, including DOC, and cold temperatures (4 $^o$C). During this season, $Al_i$ is strongly

coupled with pH, DOC, $Al_d$ and Ca in MR, but less so in MPB. A possible explanation is that season 1

is dominated by snowmelt hydrology in which cation exchange between soil and discharge occurs less

efficiently, which has been attributed to ice and frozen soil potentially limiting water contact time with

soil (Christophersen et al., 1990). The onset of season 2 (approximately late June) is characterized by

increasing $Al_i$ concentrations, temperature, and DOC. $Al_i$ and pH values are higher in this season and

$Al_i$ becomes strongly negatively correlated with pH as pH increases to the lower threshold for gibbsite.

In MR in season 2 $Al_i$ has a strong positive relationship with DOC. The highest observed $Al_i$

concentrations of the year occur in season 2 (Fig. 4). $Al_i$ relations are weak in MR in season 3

(approximately September through March), likely due to the lower frequency of measurements during

the winter. Season 3 in MR has the highest concentrations of dissolved constituents ($Al_d$, Ca, and DOC),

whereas in MPB only Ca has the highest concentrations.





### 3.4 Ecological implications

While the summer peak in $Al_i$ that we observed in $NS_A$ does not coincide with the smoltification period, when salmon transition from parr to smolt and are highly sensitive to Al exposure (Kroglund et al., 2007, Monette and McCormick, 2008, Nilsen et al., 2013), continued exposure throughout the year may still negatively affect salmon populations, as accumulation of $Al_i$ on gills reduces salmon marine and freshwater survival (Kroglund et al., 2007). Further, $Al_i$ concentrations as low as 20 ug $L^{-1}$ may impair marine survival without reducing freshwater survival (Kroglund and Staurnes, 1999, Staurnes et al., 1996), contributing to the observation that marine threats are driving population declines of Atlantic Salmon (e.g. Gibson et al., 2011). In addition, as the higher $Al_i$ concentrations appear to be driven – at least in part – by lower flow in the summer months, increases in the length and severity of droughts and heat-waves due to climate change may further increase $Al_i$ concentrations and exacerbate $Al_i$ effects on aquatic life. Increases in Al have already been observed across areas previously affected by freshwater acidification (Sterling et al., in prep.).

For example, because many peak $Al_i$ concentrations occur on the first exceptionally warm day in late spring, the peaks may be exacerbated with springtime warming associated with climate change. As warm days begin to occur earlier in the season, there may be increasing chance of the peak $Al_i$ concentrations overlapping with smoltification season and emergence of salmon fry; both considered the most vulnerable life stages of Atlantic salmon (e.g., Farmer, 2000), although the phenology of the smolt run is expected to similarly advance earlier in the year.



## 4 Conclusions

Our study reveals that widespread and persistent toxic concentrations of $Al_i$ in $NS_A$ freshwaters

pose a risk to aquatic, and potentially terrestrial, life. Previously, high DOC concentrations were

presumed to protect aquatic life against $Al_i$; our study shows that this presumption does not hold.

Our results suggest that the recent 88 to 99% population decline of the Southern Uplands

Atlantic salmon population in $NS_A$ (Gibson et al., 2011) may be partially attributable to $Al_i$, in contrast

to earlier studies which downplayed the role of $Al_i$ in Atlantic salmon mortality (Bowlby et al., 2013,

Lacroix and Townsend, 1987). These high $Al_i$ concentrations in $NS_A$ highlight the need to increase our

understanding of the influence of $Al_i$ on both terrestrial and aquatic ecosystems, and its implications for

biodiversity.

The catchments with the highest $Al_i$ levels had particularly low Ca levels, raising concerns as

Ca is protective against $Al_i$ toxicity, and highlighting coincident threats of Ca depletion and elevated

Al. Recent work has identified globally widespread low levels and declines in Ca (Weyhenmeyer et al.,

2019), raising the question of what other regions may also have $Al_i$ levels exceeding toxic thresholds.

The serious potential consequences $Al_i$ highlight the importance for actions to further reduce

acid emissions and deposition, as critical loads are still exceeded across the province (Keys, 2015), and

to adapt forest management practices to avoid base cation removal and depletion. Addition of base

cations through liming and enhanced weathering of soils and freshwaters may accelerate recovery from

acidification.



# Data availability

Readers can access our data from HydroShare supported by CUASHI, a FAIR-aligned data repository (https://www.re3data.org/).

# Author contribution

SS conceived the idea and led the writing of the MS. SM led the field data collection. SM and TAC designed the protocol for $Al_i$ sampling, assisted with data analysis and helped with the writing. LR performed spatial and statistical analysis, produced figures, and assisted with sample collection and draft writing. KH assisted with data analysis, figure production and editing and contributed to the draft. TAC provided information on analytical and field sampling methods, and selection of sampling sites. EAH contributed field samples, assisted with data analysis and contributions to the manuscript.

# Competing interests

The authors declare that they have no conflict of interest.

# Acknowledgements

The Atlantic Salmon Conservation Foundation, Atlantic Canada Opportunities Agency, the Nova Scotia Salmon Association, and Fisheries and Oceans Canada provided financial support for the field data collection and the laboratory analyses. Marley Geddes, Siobhan Takla, Franz Heubach, Lorena Heubach, Emily Bibeau and Ryan Currie provided field assistance.



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





# Tables

Table 1  Study site characteristics. "n" refers to the number of sampling events. Number in brackets after the mean concentration is the standard deviation. One

$Al_i$ outlier removed for MR (value: 2 µg·L⁻¹, date: 30 April 2015). pH is calibrated using the method outlined in Appendix D.4.





| Site | Lat | Long | Area (km²) | n | Land use | Dominant Bedrock Type | Mean Ali (ug L⁻¹) | Mean DOC (mg L⁻¹) | Mean Ald (ug L⁻¹) | Mean Ca (ug L⁻¹) | Mean pH |
|---|---|---|---|---|---|---|---|---|---|---|---|
| Mersey River (MR) Moose Pit | 44.437 | -65.223 | 292.8 | 47 | Natural forest | Granite | 22.5 (11.7) | 8.6 (2.7) | 195 (54.9) | 699 (120) | 5.1 |
| Brook (MPB) Pine Marten | 44.462 | -65.048 | 15.8 | 39 | Natural forest | Granite/slate | 20.8 (12.2) | 15.8 (6.1) | 249 (85.9) | 826 (344) | 5.0 |
| Brook (PMB) | 44.436 | -65.209 | 1.5 | 15 | Natural forest | Slate | 13.5 (12.0) | 8.6 (3.3) | 149 (43.4) | 969 (536) | 5.1 |
| Maria Brook (MB) | 44.779 | -64.414 | 0.2 | 12 | Natural forest | Granite | 40.1 (23.2) | 9.8 (4.4) | 319 (99.2) | 1292 (286) | 5.1 |
| Brandon Lake Brook (BLB) | 45.021 | -62.690 | 1.3 | 22 | Natural forest | Sandstone/slate | 48.7 (27.6) | 16.0 (8.3) | 350 (71.0) | 836 (272) | 4.9 |
| Upstream of West River Lime Doser (ALD) | 45.054 | -62.800 | 32.3 | 22 | Natural forest | Sandstone/slate | 45.3 (26.7) | 13.8 (3.7) | 243 (64.8) | 759 (126) | 5.2 |
| Upper Killag River (UKR) | 45.064 | -62.705 | 36.8 | 18 | Natural forest | Sandstone/slate | 43.5 (23.5) | 12.8 (3.0) | 224 (68.3) | 739 (230) | 5.3 |
| Little River (LR) | 44.952 | -62.611 | 47.1 | 13 | Natural forest | Sandstone/slate | 15.1 (11.7) | 7.2 (1.9) | 109 (46.1) | 746 (166) | 5.4 |
| Keef Brook (KB) | 45.0284 | -62.7153 | 2.3 | 5 | Natural forest | Sandstone/slate | 28.2 (11.5) | 10.8 (3.6) | 281 (80.4) | 621 (275) | 5.1 |
| Colwell Creek (CC) | 45.0279 | -62.7127 | 1.7 | 8 | Natural forest | Sandstone/slate | 58.9 (41.7) | 23.1 (5.1) | 411 (117) | 750 (568) | 5.0 |





Table 2. $Al_i$ relations with other stream chemistry parameters separated by possible seasons. Dark shading

represents $r^2 > 0.6$. Medium shading represents $r^2$ 0.2-0.6. Light shading represents $r^2$ 0.0-0.2. Green

indicates negative relation. Orange indicates positive relation.

| | pH | | DOC | | Tw | | $Al_d$ | | Ca | |
|---|---|---|---|---|---|---|---|---|---|---|
| | slope | $r^2$ | slope | $r^2$ | slope | $r^2$ | slope | $r^2$ | slope | $r^2$ |
| Season 1 | | | | | | | | | | |
| MR | -7.67 | 0.78 | 1.78 | 0.49 | -0.26 | 0.42 | 0.084 | 0.67 | 0.0329 | 0.50 |
| MPB | 8.44 | 0.0045 | 2.62 | 0.71 | 2.66 | 0.72 | 0.13 | 0.68 | 0.053 | 0.59 |
| Season 2 | | | | | | | | | | |
| MR | -53.2 | 0.27 | 7.5 | 0.51 | 0.72 | 0.034 | 0.23 | 0.52 | 0.13 | 0.37 |
| MPB | -19.6 | 0.22 | 1.4 | 0.43 | 1.43 | 0.23 | 0.1 | 0.42 | 0.039 | 0.42 |
| Season 3 | | | | | | | | | | |
| MR | 4.57 | 0.046 | 0.089 | 0.0014 | 0.25 | 0.088 | 0.021 | 0.014 | 0.006 | 0.0001 |
| MPB | -39.6 | 0.56 | 1.33 | 0.73 | -2.42 | 0.44 | 0.086 | 0.66 | 0.018 | 0.49 |





# Figures

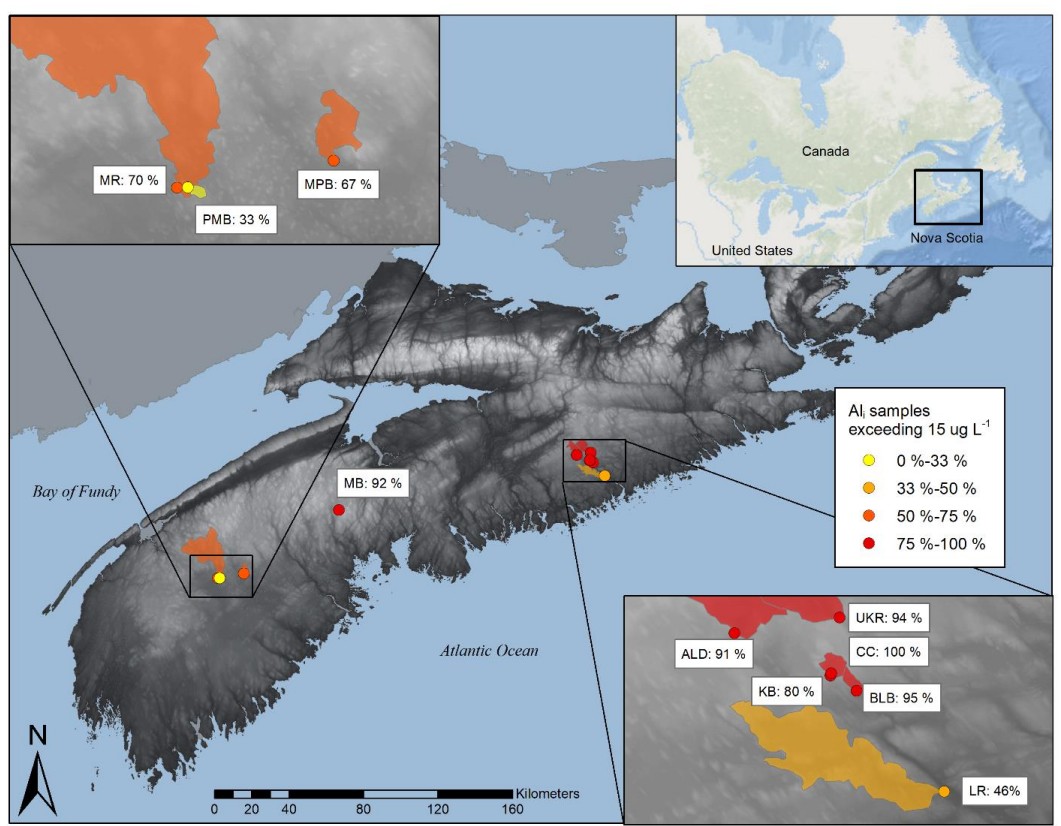

Figure 1. Study site locations showing proportion of samples when $Al_i$ concentrations exceeded the 15 μg

$L^{-1}$ toxic threshold. For additional site details, refer to Table 1.



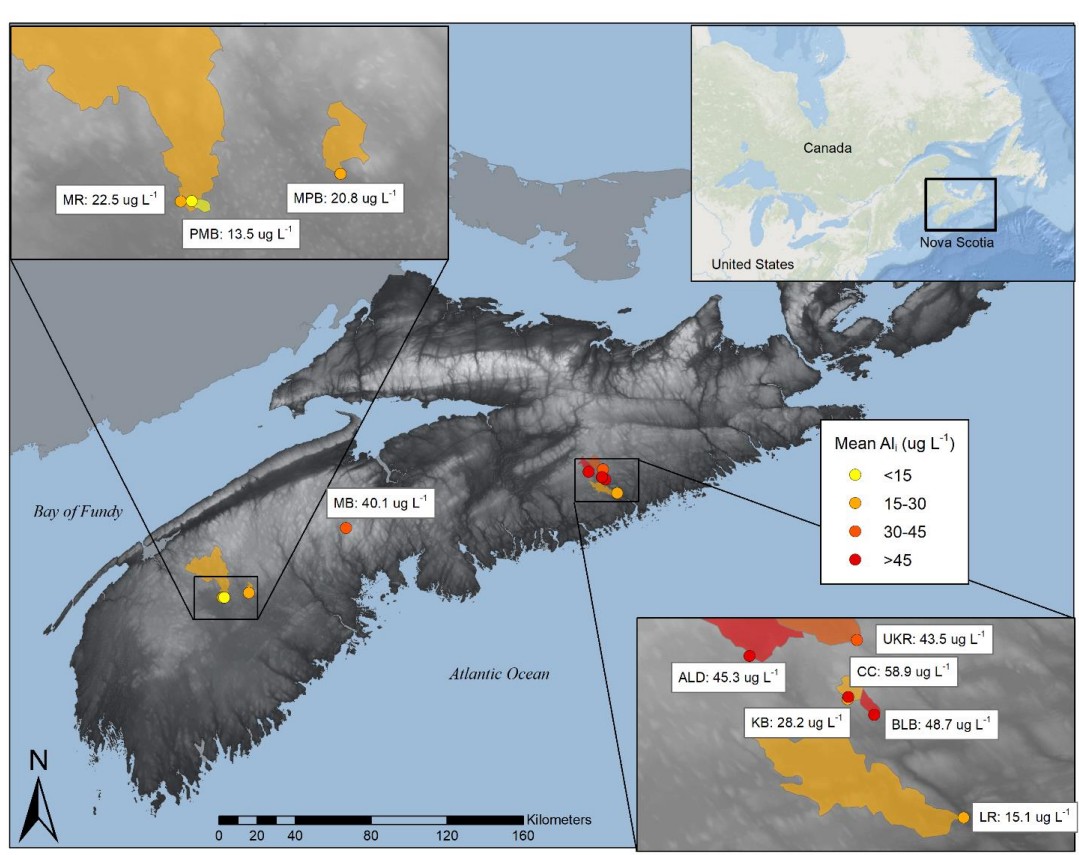

Figure 2.  Mean Al$_i$ concentrations between spring 2015 to fall 2018.





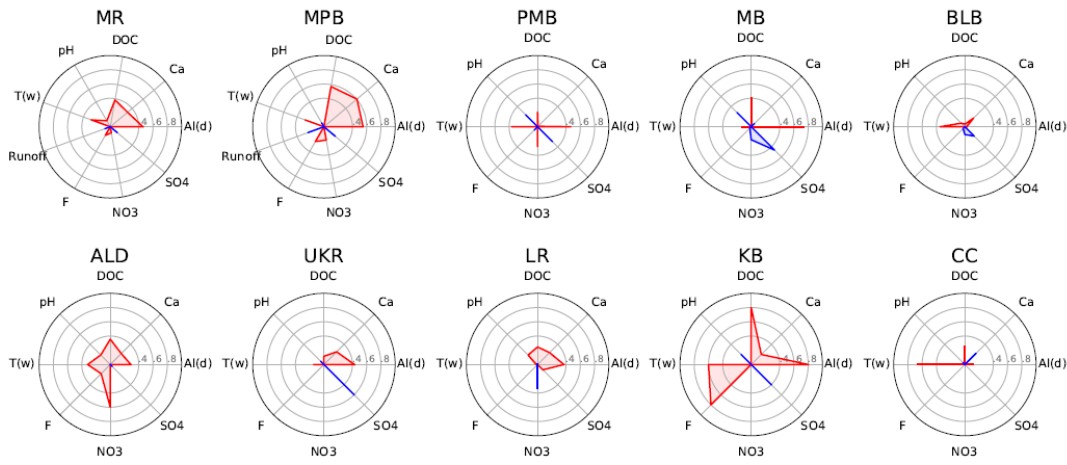

Figure 3. Correlation among water chemistry parameters and $Al_i$ concentration, where red polygons and

lines indicate a positive correlation with $Al_i$, and blue polygons and lines indicate a negative correlation





with $Al_i$. One $Al_i$ outlier removed for MR (value: 2 µg L$^{-1}$, date: 30 April 2015). Correlation data are listed

in Table A4.

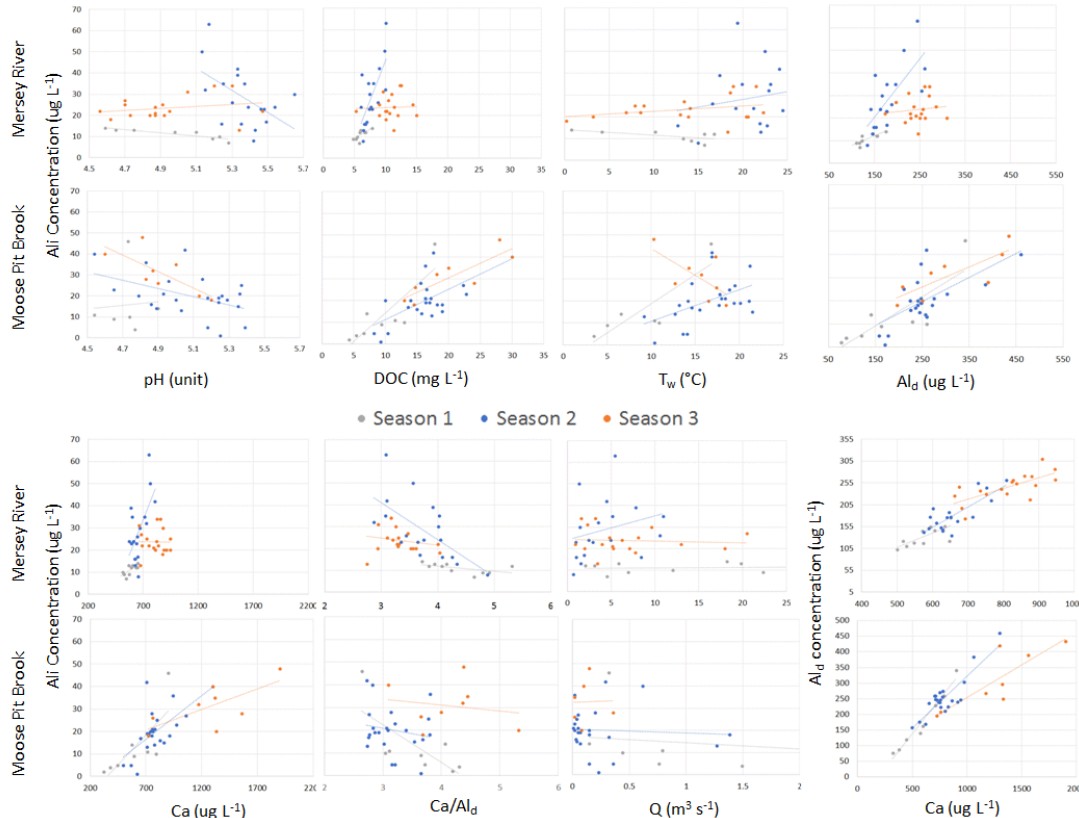

Figure 4. Scatterplot relationships among water chemistry parameters for seasons 1, 2, and 3 at MR and

MPB. $R^2$ values are listed in Table A5. One runoff outlier removed for MR (value: 17.294 m3 s$^{-1}$, date: 22

April 2015). One runoff outlier removed for MPB (value: 34.994 m3 s$^{-1}$, date: 22 April 2015).

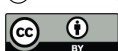

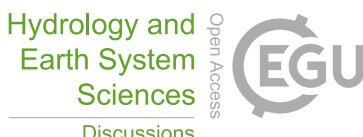
# Appendix A. Tables

Table A1 $Al_i$ terminology, speciation methodology, and trends from published studies. Several methods do not measure $Al_i$ in situ, which can cause error due to changes in temperature, DOC and pH, which vary during transit to the lab. Increased pH and increased temperature in lab conditions can cause the underestimation of $Al_i$. $Al_{nl}$=non-labile Al, $Al_m$=monomeric Al, $Al_{tm}$=total monomeric Al, $Al_{om}$=organic monomeric Al, $Al_{tr}$=total reactive Al, $Al_{nlm}$=non-labile monomeric Al, CEC= Cation Exchange Column, ICP-AES= Inductively Coupled Plasma-Atomic Emission Spectroscopy. AWMN= Acid Waters Monitoring Network.

| Al Species | Definition | Analysis Method | Trend | Location | Reference |
|---|---|---|---|---|---|
| $Al_i$ | Inorganic Al | Colourimetry ($Al_i$-$Al_{nl}$) | Decreasing $Al_i$ from 1988-2008 | AWMN in UK | Monteith et al. (2014) |
| $Al_{im}$ | Inorganic monomeric Al | Colourimetry ($Al_{tm}$-$Al_{om}$) | Decreasing $Al_i$ from 2001-2011 | New York, USA | Josephson et al. (2014) |
| $Al_i$ | Ionic Al | CEC ($Al_r$-$Al_o$) | Mean NS $Al_i$=25.3 µg/L Mean NB $Al_i$=31.0 µg/L | Atlantic Canada | Dennis and Clair (2012) |
| $Al_i$ | Ionic Al | Colourimetry | Decreasing $Al_i$ in lakes | Norway | Hesthagen et al. (2011) |
| LA1 | Inorganic Al (sum of inorganic and monomeric Al species) | ICP-AES, Flow injection, Pyrocatechol violet, and CEC ($Al_{tr}$-$Al_{nl}$) | 15% of LA1 samples were >10 µg/L | Norway | Kristensen et al. (2009) |
| Al-l | Labile/cationic/inorganic monomeric Al | Colourimetry ($Al_{tm}$-$Al_{nlm}$) | Decreasing Al-l across the UK | AWMN in UK | Evans & Monteith (2001) |
| $Al_{lim}$ | Labile Al (free and inorganically complexed Al) | Van Benschoten method | Mean $Al_{lim}$ of 72 µg/L from 2009-2010 | China | Wang et al. (2013) |





| | | | | | |
|---|---|---|---|---|---|
| $Al_i$ | Inorganic monomeric | Colourimetry and CEC $(Al_m\text{-}Al_o)$ | $Al_i$ fraction decreased in catchments between 1991 & 2007 | Czech Republic | Kram et al. (2009) |
| $Al_i$ | Inorganic Al | AAS | Decreasing $Al_i$ from 1990-2010 | Adirondack Mountains, USA | Strock et al. (2014) |


Table A2 Raw sample data. RL: rising limb of hydrograph, FL: falling limb of hydrograph, and BF: base flow. Air temperature ($T_a$) data were collected from the Kejimkujik 1 weather station (Climate ID: 8202592; 44.24'11.020 ºN, 65.12'11.070 ºW) for MR, MPB, PMB, and MB, and the Stanfield Airport weather station (Climate ID: 8202251; 44°52'52.000" N, 63°30'31.000" W) for CC, KB, ALD, BLB, UKR, and LR. Missing $T_a$ data were replaced with data from another local meteorological tower located one kilometer to the northwest of the MPB site (44.469549, -65.061295).

| Site | Date | Ali (µg L$^{-1}$) | Alo/Ala (%) | Season | Ala (µg L$^{-1}$) | Ca (µg L$^{-1}$) | DOC (mg L$^{-1}$) | SO$_4$ (µg L$^{-1}$) | pH (unit) | Tw(°C) | Ta (°C) | Discharge (m³ s$^{-1}$) | Runoff (mm day$^{-1}$) | Hydrograph Stage |
|---|---|---|---|---|---|---|---|---|---|---|---|---|---|---|
| ALD | 2016-04-29 | 19 | 87.7 | 1 | 155 | 591 | 7 | 899 | 4.67 | 6.8 | 4 | | | |
| ALD | 2016-05-19 | 12 | 94.1 | | 202 | 800 | 10.7 | 1414 | 5.89 | | 12.0 | | | |
| ALD | 2016-06-03 | 25 | 90.7 | 2 | 268 | 722 | 12.5 | 639 | 5.02 | 16.6 | 13.2 | | | |
| ALD | 2016-06-16 | 32 | 88.3 | 2 | 274 | 674 | 12.9 | 578 | 4.99 | 13.2 | 13 | | | |
| ALD | 2016-06-28 | 28 | 89.4 | 2 | 265 | 720 | 12.2 | 959 | 5.26 | 22.1 | 24.2 | | | |
| ALD | 2016-07-15 | 37 | 87 | 2 | 285 | 792 | 15 | 761 | 5.11 | 20.7 | 19.6 | | | |
| ALD | 2016-08-05 | 48 | 79.9 | | 239 | 700 | 19.4 | 1414 | 5.98 | | 21.2 | | | |
| ALD | 2016-09-10 | 48 | 78.2 | | 220 | 1000 | 14.8 | 2000 | 5.03 | | 20.8 | | | |
| ALD | 2016-10-02 | 13 | 92.3 | | 169 | 1000 | 14.4 | 3000 | 5.27 | | 11.4 | | | |
| ALD | 2016-11-19 | 44 | 82 | | 245 | 900 | 14.6 | 1414 | 5.03 | | 7.6 | | | |
| ALD | 2017-04-19 | 27 | 81.1 | 1 | 143 | 600 | 10.4 | 1209 | 4.55 | 7.8 | 3.2 | | | |
| ALD | 2017-05-14 | 69 | 61 | 2 | 177 | 600 | 12.1 | 923 | 4.92 | 13.4 | 4 | | | |
| ALD | 2017-05-30 | 37 | 85.8 | 2 | 261 | 600 | 11.8 | 2536 | 4.77 | 14.3 | 12.2 | | | |
| ALD | 2017-06-22 | 100 | 59.8 | 2 | 249 | 700 | 15.2 | 1414 | 5.17 | 22.8 | 25 | | | |





| | | | | | | | | | | | |
|---|---|---|---|---|---|---|---|---|---|---|---|
| ALD | 2017-07-13 | 62 | 80.3 | 2 | 315 | 800 | 19.3 | 1414 | 5.24 | 20.6 | 18.5 |
| ALD | 2017-08-01 | 26 | 89 | 2 | 236 | 800 | 15.1 | 1414 | 4.96 | 25.6 | 28.4 |
| ALD | 2017-08-23 | 35 | 84.4 | 2 | 224 | 700 | 13.2 | 1125 | 5.14 | 21.8 | 21 |
| ALD | 2017-09-16 | 77 | 82.5 | 2 | 439 | 1000 | 23.5 | | 4.73 | 20.7 | 18.7 |
| ALD | 2018-05-10 | 46 | 75.7 | | 189 | 700 | 8.8 | 1414 | 5.64 | | 7.5 |
| ALD | 2018-06-07 | 43 | 83.8 | | 266 | 700 | 16.1 | 1414 | 5.13 | | 11.0 |
| ALD | 2018-07-05 | 119 | 62.5 | | 317 | 800 | 13.6 | 1414 | 5.61 | | 23.8 |
| ALD | 2018-11-23 | 50 | 76 | | 208 | 800 | 10.1 | 1414 | 5.45 | | -9.1 |
| BLB | 2016-04-29 | 20 | 89.5 | 2 | 190 | 476 | 7.2 | 936 | 5.03 | 5.7 | 4 |
| BLB | 2016-06-03 | 60 | 82.1 | 4 | 336 | 770 | 11.9 | 669 | 4.78 | 10.1 | 13.5 |
| BLB | 2016-06-16 | 33 | 91.2 | 4 | 373 | 789 | 13.2 | 1158 | 4.77 | 9.8 | 13 |
| BLB | 2016-06-28 | 26 | 93.3 | 4 | 388 | 894 | 13.6 | 1251 | 4.67 | 13.1 | 23.9 |
| BLB | 2016-07-15 | 42 | 90.5 | 4 | 443 | 887 | 16.7 | 723 | 4.77 | 14.3 | 18.7 |
| BLB | 2016-08-05 | 6 | 98.6 | | 429 | 1000 | 26.2 | 1414 | 5.29 | | 21.2 |
| BLB | 2016-09-10 | 81 | 77.1 | | 354 | 900 | 48.3 | 1414 | 4.87 | | 20.8 |
| BLB | 2016-10-02 | 33 | 90.1 | | 335 | 1000 | 18.5 | 2000 | 5.1 | | 11.4 |
| BLB | 2016-11-19 | 28 | 92.6 | | 379 | 1000 | 17.2 | 1414 | 4.76 | | 7.6 |
| BLB | 2017-04-19 | 41 | 79.1 | 4 | 196 | 600 | 9.6 | 1927 | | 4.2 | 4 |
| BLB | 2017-05-14 | 46 | 82.6 | 4 | 264 | 800 | 12.9 | 1550 | | 7.7 | 6 |
| BLB | 2017-05-30 | 36 | 88.3 | 4 | 308 | 700 | 11.3 | 1795 | | 8.4 | 14.9 |
| BLB | 2017-06-22 | 110 | 70.1 | 4 | 368 | 800 | 14.9 | 1414 | 4.8 | 17.3 | 24.6 |





| Site | Date | | | | | | | | | | |
|------|------|------|------|------|------|------|------|------|------|------|------|
| BLB | 2017-07-13 | 50 | 88.3 | 4 | 427 | 900 | 17.6 | 1414 | 4.87 | 15.8 | 17 |
| BLB | 2017-08-01 | 37 | 90.7 | 4 | 396 | 800 | 17.9 | 1414 | 4.7 | 20.6 | 29 |
| BLB | 2017-08-23 | 54 | 85.8 | 3 | 381 | 1000 | 17.1 | 1172 | 4.94 | 18.3 | 21 |
| BLB | 2017-09-16 | 34 | 91.9 | 4 | 420 | 1000 | 17.3 | | 4.52 | 16.6 | 18.9 |
| BLB | 2018-05-10 | 37 | 85.5 | | 256 | 700 | 8.5 | 1414 | 5.16 | | 7.5 |
| BLB | 2018-06-07 | 86 | 75 | | 344 | 800 | 15.7 | 1414 | 5.29 | | 11.0 |
| BLB | 2018-07-05 | 83 | 80.3 | | 421 | 900 | 13.8 | 1414 | 5.42 | | 23.8 |
| BLB | 2018-10-02 | 104 | 67.4 | | 319 | 1600 | 12.4 | 1414 | 5.04 | | 7.7 |
| BLB | 2018-11-23 | 24 | 93.5 | | 367 | 70.7 | 10.5 | 1414 | 4.8 | | -9.1 |
| CC | 2016-06-03 | 32 | 91.9 | 4 | 397 | 501 | 15.2 | 385 | 4.66 | 11.2 | 13.5 |
| CC | 2016-06-16 | 46 | 88.9 | 4 | 413 | 520 | 17.7 | 304 | 4.71 | 10.4 | 12.8 |
| CC | 2016-06-28 | 107 | 78.9 | 4 | 507 | 537 | 21 | 401 | 4.82 | 14.8 | 24.2 |
| CC | 2016-07-15 | 53 | 89.9 | 4 | 524 | 642 | 26 | 208 | 4.6 | 14.6 | 18.7 |
| CC | 2016-08-05 | 140 | 68.6 | | 446 | 400 | 29.3 | 1414 | 5.73 | | 21.2 |
| CC | 2016-09-10 | 32 | 86.9 | | 244 | 400 | 22.2 | 1414 | 4.72 | | 20.8 |
| CC | 2016-10-02 | 34 | 85.5 | | 234 | 900 | 28.8 | 1414 | 4.95 | | 11.4 |
| CC | 2016-11-19 | 27 | 94.9 | | 527 | 2100 | 24.9 | 1414 | 6.11 | | 7.6 |
| KB | 2016-04-29 | 14 | 90.6 | 2 | 149 | 1110 | 5.7 | 1061 | 5.69 | 8.2 | 4 |
| KB | 2016-06-03 | 20 | 92.5 | 2 | 267 | 459 | 9.9 | 611 | 4.89 | 14.1 | 13.5 |
| KB | 2016-06-16 | 38 | 87.7 | 2 | 310 | 515 | 11.3 | 852 | 4.9 | 12.3 | 10.8 |
| KB | 2016-06-28 | 28 | 91.3 | 2 | 323 | 486 | 11.7 | 887 | 5.06 | 17.8 | 24.5 |
| KB | 2016-07-15 | 41 | 88.5 | 2 | 356 | 535 | 15.6 | 621 | 5.03 | 18.7 | 18.7 |





| | | | | | | | | | | | |
|---|---|---|---|---|---|---|---|---|---|---|---|
| LR | 2016-08-05 | 27 | 50 | | 54 | 1100 | 5.7 | 1414 | 6.03 | | 21.2 |
| LR | 2016-09-10 | 3 | 92.1 | | 38 | 800 | 4.4 | 1414 | 6.07 | | 20.8 |
| LR | 2016-10-02 | 6 | 95.2 | | 124 | 900 | 10.1 | 2000 | 5.76 | | 11.4 |
| LR | 2017-04-19 | 4 | 96.6 | 1 | 116 | 600 | 7.1 | 1416 | 4.87 | 6 | 1.7 |
| LR | 2017-05-14 | 20 | 84.6 | 2 | 130 | 600 | 8.1 | 1213 | 4.95 | 12.3 | 6 |
| LR | 2017-05-30 | 17 | 89 | 2 | 154 | 600 | 8.6 | 1572 | 5.21 | 15 | 12.5 |
| LR | 2017-06-22 | 34 | 69.9 | 2 | 113 | 700 | 8.2 | 1414 | 5.51 | 19.6 | 19 |
| LR | 2017-07-13 | 12 | 88.7 | 2 | 106 | 600 | 6.4 | 1414 | 5.54 | 21.8 | 18 |
| LR | 2017-08-01 | 2 | 96.9 | 2 | 65 | 600 | 6.6 | 1414 | 5.1 | 19.6 | 24.8 |
| LR | 2017-08-23 | 5 | 88.4 | 2 | 43 | 600 | 4.1 | 1371 | 5.37 | 21.6 | 21.3 |
| LR | 2017-09-16 | 5 | 94.9 | 2 | 99 | 700 | 6.7 | 1414 | 5.01 | 19.4 | 15.8 |
| LR | 2018-05-10 | 35 | 74.1 | | 135 | 800 | 6.7 | 1414 | 5.54 | | 7.5 |
| LR | 2018-06-07 | 26 | 84.0 | | 162 | 900 | 8.2 | 1414 | 5.55 | | 11.0 |
| MB | 2016-05-27 | 30 | 88.9 | 2 | 270 | 1200 | 6.8 | 1278 | 5.14 | 9.8 | 12 |
| MB | 2016-06-15 | 15 | 94.2 | 2 | 260 | 1590 | 8.4 | 1497 | 5.61 | 11.2 | 14.6 |
| MB | 2016-06-27 | 27 | 90.5 | 2 | 284 | 1610 | 7.6 | 1851 | 5.28 | 16.3 | 16.7 |
| MB | 2016-07-14 | 40 | 86.9 | 2 | 305 | 1780 | 6.4 | 1747 | 5.4 | 15.5 | 28.5 |
| MB | 2017-04-20 | 25 | 89.8 | 1 | 246 | 848 | 7 | 1996 | 4.86 | 2.3 | 4 |
| MB | 2017-05-13 | 48 | 84.1 | 1 | 302 | 977 | 7.2 | 1385 | 4.76 | 9.1 | 17 |
| MB | 2017-05-29 | 40 | 87.9 | 2 | 330 | 1100 | 9 | 1977 | 4.99 | 9.1 | 14.5 |
| MB | 2017-06-21 | 96 | 81.2 | 2 | 510 | 1480 | 15.8 | 551 | 5.18 | 13.7 | 23.3 |
| MB | 2017-07-12 | 46 | 87.7 | 2 | 375 | 1320 | 11.5 | 28968 | 5.13 | 15.8 | 25.9 |





| | | | | | | | | | | | | | | |
|---|---|---|---|---|---|---|---|---|---|---|---|---|---|---|
| MB | 2017-07-31 | 43 | 87.7 | 2 | 351 | 1470 | 12.1 | 1629 | 5.08 | 15.6 | 27.4 | | | |
| MB | 2017-08-22 | 80 | 85.7 | 2 | 560 | 1500 | 21 | 828 | 4.91 | 15.5 | 27.6 | | | |
| MB | 2017-09-17 | 30 | 89.3 | 3 | 280 | 1600 | 11 | 1258 | 5.14 | 14.7 | 23 | | | |
| MPB | 2015-04-22 | 2 | 97.1 | 1 | 77 | 323 | 4.3 | 1009 | | | 7.3 | 6.41 | 34.992 | RL |
| MPB | 2015-04-30 | 4 | 95.9 | 1 | 88 | 379 | 5.4 | 1272 | 4.77 | 3.5 | 4.5 | 1.49 | 8.134 | FL |
| MPB | 2015-05-06 | 5 | 95.8 | 1 | 120 | 446 | 6.6 | 1304 | | | 14 | 0.76 | 4.149 | BF |
| MPB | 2015-05-13 | 5 | 96.8 | 2 | 158 | 498 | 8.2 | 958 | 5.18 | 13.6 | 7 | 0.36 | 1.965 | RL |
| MPB | 2015-05-20 | 1 | 99.4 | 2 | 170 | 621 | 9.3 | 815 | 5.25 | 10.4 | 12 | 0.23 | 1.256 | RL |
| MPB | 2015-05-27 | 5 | 97.2 | 2 | 177 | 567 | 10.4 | 699 | 5.39 | 14.1 | 21 | 0.15 | 0.819 | RL |
| MPB | 2015-06-03 | 13 | 95 | 2 | 260 | 710 | 17.3 | 639 | 5.03 | 9.2 | 8 | 1.27 | 6.933 | RL |
| MPB | 2015-06-10 | 17 | 92.8 | 2 | 236 | 651 | 13.6 | 443 | 5.24 | 14.6 | 10 | 0.32 | 1.747 | RL |
| MPB | 2015-06-17 | 28 | 88.3 | 2 | 239 | 751 | 15.6 | 560 | 5.15 | 14.6 | 16 | 0.2 | 1.092 | RL |
| MPB | 2015-06-24 | 18 | 93.4 | 2 | 271 | 751 | 19 | 357 | 5 | 13.2 | 18 | 1.38 | 7.533 | RL |
| MPB | 2015-07-02 | 42 | 83.8 | 2 | 259 | 705 | 17.6 | 322 | 5.05 | 16.9 | 20 | 0.29 | 1.583 | BF |
| MPB | 2015-07-08 | 19 | 92.3 | 2 | 247 | 724 | 16.4 | 400 | 5.24 | 19.4 | 23 | 0.07 | 0.382 | BF |
| MPB | 2015-07-15 | 19 | 92.3 | 2 | 248 | 710 | 17 | 464 | 5.18 | 20.1 | 18 | 0.05 | 0.273 | BF |
| MPB | 2015-07-22 | 21 | 91.5 | 2 | 247 | 756 | 16.3 | 552 | 5.36 | 18.4 | 17 | 0.05 | 0.273 | RL |
| MPB | 2015-07-29 | 18 | 92.5 | 2 | 240 | 912 | 18.2 | 1146 | 5.29 | 17.7 | 19 | 0.15 | 0.819 | RL |
| MPB | 2015-08-05 | 15 | 93.9 | 2 | 244 | 863 | 19 | 650 | 5.35 | 21.5 | 19 | 0.04 | 0.218 | FL |
| MPB | 2015-08-12 | 25 | 88.2 | 2 | 211 | 798 | 16.5 | 618 | 5.37 | 18.9 | 21 | 0.04 | 0.218 | RL |
| MPB | 2015-08-19 | 36 | 85.4 | 2 | 247 | 941 | 16.3 | 721 | 4.83 | 21.2 | 24 | 0.02 | 0.109 | BF |
| MPB | 2015-08-26 | 20 | 91.1 | 2 | 224 | 761 | 10 | 607 | 5.26 | 21.1 | 16 | 0.02 | 0.109 | BF |



| Site | Date | | | | | | | | | | | | | |
|---|---|---|---|---|---|---|---|---|---|---|---|---|---|---|
| MPB | 2015-09-02 | 26 | 87.5 | 3 | 208 | 760 | 14.7 | 711 | 4.9 | 17.4 | 21 | 0.02 | 0.109 | BF |
| MPB | 2015-09-09 | 18 | 90.8 | 3 | 196 | 722 | 14.5 | 823 | 5.2 | 18.5 | 20 | | 0 | RL |
| MPB | 2015-09-16 | 20 | 92 | 3 | 250 | 1330 | 13 | 4375 | 5.13 | 16.5 | 19 | 0.08 | 0.437 | BF |
| MPB | 2015-09-23 | 35 | 88.2 | 3 | 297 | 1320 | 20 | 2598 | 5 | 14.3 | 17 | 0.02 | 0.109 | BF |
| MPB | 2015-09-30 | 32 | 88.1 | 3 | 268 | 1170 | 18.1 | 1902 | 4.87 | 15.7 | 19 | | 0 | BF |
| MPB | 2015-10-07 | 48 | 88.9 | 3 | 434 | 1900 | 28 | 2576 | 4.81 | 10.3 | 13 | 0.15 | 0.819 | BF |
| MPB | 2015-10-14 | 28 | 92.8 | 3 | 390 | 1560 | 24 | 1963 | 4.83 | 12.7 | 16 | 0.36 | 1.965 | RL |
| MPB | 2016-04-28 | 14 | 90.1 | 1 | 141 | 573 | 7.1 | 800 | 4.9 | 6.6 | 4 | 0.15 | 0.819 | RL |
| MPB | 2016-05-27 | 20 | 91.7 | 2 | 240 | 740 | 14 | 489 | 4.79 | 14.2 | 12 | 0.15 | 0.819 | RL |
| MPB | 2016-06-15 | 14 | 94.6 | 2 | 257 | 775 | 15.7 | 478 | 4.89 | 12.7 | 14.1 | 0.07 | 0.382 | FL |
| MPB | 2016-06-27 | 21 | 92.4 | 2 | 275 | 778 | 17.2 | 587 | 4.93 | 18 | 27 | 0.01 | 0.055 | FL |
| MPB | 2016-07-14 | 16 | 92.9 | 2 | 225 | 828 | 15 | 1447 | 4.86 | 15.5 | 20 | 0.03 | 0.164 | FL |
| MPB | 2017-04-20 | 9 | 94.5 | 1 | 163 | 595 | 9.4 | 1625 | 4.65 | 5 | 1 | | 0 | |
| MPB | 2017-05-13 | 11 | 95.2 | 1 | 229 | 712 | 11.5 | 1430 | 4.54 | 10.4 | 17 | 0.79 | 4.313 | FL |
| MPB | 2017-05-29 | 10 | 96.2 | 1 | 260 | 790 | 13 | 1567 | 4.74 | 10.9 | 12 | 0.44 | 2.402 | FL |
| MPB | 2017-06-21 | 46 | 86.5 | 1 | 341 | 901 | 17.8 | 226 | 4.73 | 16.8 | 24.2 | 0.32 | 1.747 | FL |
| MPB | 2017-07-12 | 27 | 93 | 2 | 384 | 1060 | 22.3 | 229 | 4.96 | 19.5 | 25.9 | 0.05 | 0.273 | FL |
| MPB | 2017-07-31 | 23 | 92.4 | 2 | 303 | 972 | 22.8 | 724 | 4.65 | 17.8 | 27 | 0.02 | 0.109 | FL |
| MPB | 2017-08-22 | 40 | 91.3 | 2 | 460 | 1300 | 30 | 255 | 4.54 | 16.9 | 28.4 | 0.62 | 3.385 | FL |
| MPB | 2017-09-17 | 40 | 90.5 | 3 | 420 | 1300 | 30 | 301 | 4.6 | 17.3 | 20.1 | 0.1 | 0.546 | FL |
| MR | 2015-04-22 | 12 | 90.2 | 1 | 122 | 648 | 5.9 | 1321 | | | 7.3 | 58.61 | 1.837 | RL |
| MR | 2015-04-30 | 2 | 98 | 1 | 102 | 500 | 5.6 | 1189 | 5 | 4.2 | 4.5 | 33.03 | 1.454 | FL |




| | | | | | | | | | | | | | | |
|---|---|---|---|---|---|---|---|---|---|---|---|---|---|---|
| MR | 2015-05-06 | 9 | 91.8 | 1 | 110 | 527 | 4.8 | 1112 | | | 14 | 22.33 | 1.269 | BF |
| MR | 2015-05-13 | 10 | 91.8 | 1 | 122 | 517 | 5.5 | 1117 | 5.23 | 13.3 | 7 | 12.05 | 1.048 | FL |
| MR | 2015-05-20 | 9 | 92.3 | 1 | 117 | 574 | 5.3 | 1101 | 5.19 | 14.2 | 12 | 6.95 | 0.912 | FL |
| MR | 2015-05-27 | 7 | 94.1 | 1 | 118 | 548 | 5.8 | 1161 | 5.28 | 15.7 | 21 | 4.53 | 0.835 | FL |
| MR | 2015-06-03 | 16 | 89.2 | 2 | 148 | 629 | 6.6 | 1069 | 5.35 | 12.7 | 8 | 8.42 | 0.946 | RL |
| MR | 2015-06-10 | 39 | 74.2 | 2 | 151 | 590 | 6.2 | 1220 | 5.33 | 17.4 | 10 | 7.8 | 0.934 | RL |
| MR | 2015-06-17 | 24 | 83.1 | | 142 | 575 | 6.1 | 1175 | 5.39 | 19.2 | | 4.98 | 0.858 | |
| MR | 2015-06-24 | 26 | 86.2 | 2 | 188 | 647 | 8.8 | 968 | 5.3 | 16.6 | 20 | 10.58 | 1.028 | BF |
| MR | 2015-07-02 | 35 | 82.1 | 2 | 196 | 602 | 8.1 | 897 | 5.25 | 19.9 | 23 | 10.94 | 1.018 | BF |
| MR | 2015-07-08 | 35 | 80.2 | 2 | 177 | 713 | 7.3 | 972 | 5.37 | 23.1 | 18 | 5.14 | 0.864 | BF |
| MR | 2015-07-15 | 23 | 87 | 2 | 177 | 593 | 7.9 | 959 | 5.46 | 24.5 | 17 | 2.9 | 0.76 | BF |
| MR | 2015-07-22 | 17 | 90.4 | 2 | 177 | 652 | 7 | 1011 | 5.49 | 21.9 | 19 | 1.9 | 0.701 | RL |
| MR | 2015-07-29 | 24 | 85.3 | 2 | 163 | 611 | 7.7 | 1146 | 5.54 | 21.2 | 19 | 2.45 | 0.735 | FL |
| MR | 2015-08-05 | 30 | 82 | 2 | 167 | 670 | 7.5 | 1077 | 5.65 | 25.2 | 21 | 1.46 | 0.671 | RL |
| MR | 2015-08-12 | 13 | 91 | 2 | 145 | 629 | 6.5 | 1094 | 5.43 | 22 | 24 | 1.53 | 0.686 | BF |
| MR | 2015-08-19 | 23 | 86.9 | 2 | 176 | 641 | 7.4 | 1097 | 5.48 | 25.3 | 16 | 0.96 | 0.632 | BF |
| MR | 2015-08-26 | 42 | 83.9 | 2 | 261 | 808 | 9 | 1179 | 5.33 | 24.1 | 21 | 4.47 | 0.731 | BF |
| MR | 2015-09-02 | 34 | 87.5 | 3 | 271 | 859 | 12.3 | 1168 | 5.3 | 21.5 | 20 | 1.59 | 0.681 | BF |
| MR | 2015-09-09 | 22 | 90.4 | 3 | 229 | 751 | 10.2 | 776 | 5.47 | 22.3 | 19 | 0.93 | 0.63 | BF |
| MR | 2015-09-16 | 34 | 87 | 3 | 261 | 828 | 12.5 | 1108 | 5.2 | 18.9 | 17 | 3.2 | 0.781 | BF |
| MR | 2015-09-23 | 13 | 94.7 | 3 | 246 | 675 | 11.3 | 900 | 5.34 | 18.3 | 19 | 3.44 | 0.789 | BF |
| MR | 2015-09-30 | 31 | 86.2 | 3 | 225 | 662 | 9.6 | 911 | 5.05 | 18.6 | | 2.3 | 0.733 | BF |





| Site | Date | | | | | | | | pH | | | | | Type |
|---|---|---|---|---|---|---|---|---|---|---|---|---|---|---|
| MR | 2015-10-07 | 21 | 91.3 | 3 | 241 | 794 | 10.7 | 989 | 4.87 | 13 | 13 | 5.16 | 0.869 | RL |
| MR | 2015-10-14 | 24 | 90.7 | 3 | 257 | 824 | 11.4 | 1166 | 4.87 | 14.1 | 16 | 6.26 | 0.905 | BF |
| MR | 2015-10-21 | 25 | 89.5 | 3 | 237 | 735 | 9 | 890 | 4.91 | 8.9 | 5 | 4.83 | 0.855 | FL |
| MR | 2015-10-28 | 22 | 91.3 | 3 | 253 | 837 | 10 | 1153 | 4.95 | 6.9 | 3 | 3.98 | 0.814 | RL |
| MR | 2015-11-04 | 25 | 91.3 | 3 | 286 | 945 | 14.4 | 967 | 4.7 | 7.9 | 7 | 8.1 | 0.947 | FL |
| MR | 2015-12-02 | 20 | 92.4 | 3 | 262 | 946 | 12 | 1139 | 4.73 | 3.2 | 6 | 17.96 | 1.183 | FL |
| MR | 2016-01-05 | 30 | 88.9 | 3 | 270 | 880 | 11 | 1245 |  |  | -20 | 9.62 | 0.998 | RL |
| MR | 2016-02-02 | 18 | 91.7 | 1 | 217 | 875 | 10.1 | 1290 | 4.62 | 0.2 | -3 | 7.75 | 0.926 | BF |
| MR | 2016-02-23 | 14 | 92 | 1 | 175 | 651 | 7.9 | 1316 | 4.59 | 0.8 | -6 | 18.21 | 1.2 | RL |
| MR | 2016-03-29 | 13 | 91.1 | 1 | 146 | 606 | 6.1 | 1060 | 4.65 | 4.2 | 2 | 19.81 | 1.248 | FL |
| MR | 2016-04-28 | 13 | 91 | 1 | 145 | 572 | 6 | 937 | 4.75 | 10.2 | 4 | 5.85 | 0.892 | FL |
| MR | 2016-05-27 | 12 | 92.3 | 1 | 156 | 635 | 6.8 | 922 | 4.98 | 16.8 | 12 | 3.11 | 0.81 | FL |
| MR | 2016-06-15 | 12 | 92.3 | 2 | 155 | 595 | 6.7 | 1217 | 5.1 | 15.7 | 14.4 | 2.05 | 0.773 | FL |
| MR | 2016-06-27 | 16 | 89.5 | 2 | 153 | 624 | 6.8 | 1263 | 5.24 | 22.7 | 24 | 1.04 | 0.649 | FL |
| MR | 2016-07-14 | 8 | 94 | 3 | 134 | 654 | 6.4 | 1697 | 5.42 | 15 | 16 | 0.68 | 0.635 | BF |
| MR | 2017-04-20 | 22 | 87.3 | 3 | 173 | 692 | 5.3 | 1625 | 4.56 | 8.5 | 1 | 13 |  | FL |
| MR | 2017-05-13 | 27 | 86.3 | 3 | 197 | 683 | 10.5 | 1437 | 4.7 | 13.4 | 13 | 20.5 | 1.28 | FL |
| MR | 2017-05-29 | 20 | 91.3 | 2 | 230 | 810 | 9 | 1774 | 4.87 | 13.9 | 10.4 | 7.08 | 0.905 | FL |
| MR | 2017-06-21 | 63 | 74.2 | 2 | 244 | 752 | 10.1 | 458 | 5.17 | 19.4 | 20.2 | 5.42 | 0.881 | FL |
| MR | 2017-07-12 | 32 | 87.4 | 2 | 254 | 729 | 10 | 982 | 5.15 | 22.9 | 23.9 | 3.55 | 0.813 | FL |
| MR | 2017-07-31 | 50 | 76.7 | 3 | 215 | 766 | 9.88 | 1116 | 5.13 | 22.5 | 24.9 | 1.37 | 0.665 | FL |
| MR | 2017-08-22 | 20 | 93.5 | 3 | 310 | 910 | 15 | 861 | 4.92 | 20.4 | 25.5 | 5.26 | 0.878 | FL |





| Site | Date | | | | | | | | | | | | | FL |
|------|------|---|---|---|---|---|---|---|---|---|---|---|---|----|
| MR  | 2017-09-17 | 20 | 92   | 3 | 250 | 890  | 15   | 817    | 4.84 | 20.6 | 17.3 | 1.98 | 0.715 | |
| PMB | 2015-05-27 | 2  | 98.4 | 2 | 128 | 742  | 7.2  | 845    | 5.62 | 12.6 | 21   | | | |
| PMB | 2015-06-03 | 6  | 95.7 | 2 | 138 | 586  | 8.8  | 1042   | 5.28 | 12.2 | 8    | | | |
| PMB | 2016-04-28 | 6  | 93.6 | 2 | 93  | 675  | 3.6  | 1244   | 5.25 | 8.2  | 4    | | | |
| PMB | 2016-05-27 | 35 | 78.1 | 2 | 160 | 900  | 8    | 691    | 4.93 | 12.7 | 12   | | | |
| PMB | 2016-06-15 | 5  | 96.7 | 2 | 151 | 1150 | 8.1  | 1229   | 5.14 | 10.9 | 14.2 | | | |
| PMB | 2016-06-27 | 5  | 94.3 | 2 | 82  | 1570 | 5.4  | 3167   | 5.35 | 14   | 24   | | | |
| PMB | 2016-07-14 | 10 | 89.3 | 2 | 96  | 1770 | 6.9  | 5652   | 5.4  | 15   | 12   | | | |
| PMB | 2017-04-20 | 4  | 96.5 | 1 | 114 | 71   | 5.3  | 2234   | 4.78 | 8.5  | 2    | | | |
| PMB | 2017-05-13 | 11 | 92.1 | 1 | 139 | 71   | 6.2  | 1328   | 4.69 | 9.8  | 16   | | | |
| PMB | 2017-05-29 | 10 | 93.8 | 2 | 160 | 730  | 7    | 2405   | 5.03 | 13.9 | 10.8 | | | |
| PMB | 2017-06-21 | 32 | 85.6 | 2 | 222 | 955  | 11.1 | 289    | 4.98 | 15.5 | 21.4 | | | |
| PMB | 2017-07-12 | 35 | 80.3 | 2 | 178 | 1580 | 10.7 | 1428   | 5.21 | 16   | 24.6 | | | |
| PMB | 2017-07-31 | 1  | 99.3 | 2 | 148 | 1780 | 13   | 2746   | 4.99 | 13.8 | 25.6 | | | |
| PMB | 2017-08-22 | 20 | 90.9 | 3 | 220 | 960  | 13   | 571    | 4.85 | 16.4 | 26.9 | | | |
| PMB | 2017-09-17 | 20 | 90   | 3 | 200 | 990  | 15   | 640    | 4.7  | 16   | 17.8 | | | |
| UKR | 2016-05-19 | 21 | 89.7 |   | 203 | 700  | 10.4 | 1414.2 | 5.83 |      | 12.0 | | | |
| UKR | 2016-08-05 | 18 | 88.5 |   | 157 | 700  | 15.1 | 1414.2 | 5.56 |      | 21.2 | | | |
| UKR | 2016-09-10 | 16 | 89.9 |   | 158 | 100  | 12.1 | 1414.2 | 5.58 |      | 20.8 | | | |
| UKR | 2016-10-02 | 15 | 91.8 |   | 182 | 900  | 13.8 | 1414.2 | 5.77 |      | 11.4 | | | |
| UKR | 2016-11-19 | 41 | 84.4 |   | 262 | 1100 | 15.1 | 2000   | 4.89 |      | 7.6  | | | |
| UKR | 2017-04-19 | 38 | 72.3 | 3 | 137 | 500  | 9.5  | 1292   |      | 7.3  | 3.4  | | | |


| | | | | | | | | | | | | |
|---|---|---|---|---|---|---|---|---|---|---|---|---|
| UKR | 2017-05-14 | 24 | 87.2 | 2 | 187 | 600 | 12.6 | 1049 | | 12.9 | 6 |
| UKR | 2017-05-30 | 37 | 83.3 | 2 | 221 | 600 | 9.8 | 1115 | | 15.2 | 12.5 |
| UKR | 2017-06-22 | 66 | 67.5 | 2 | 203 | 800 | 12.1 | 1414 | 5.22 | 23.4 | 24.2 |
| UKR | 2017-07-13 | 47 | 85.4 | 2 | 322 | 800 | 17.6 | 1414 | 5.21 | 22.3 | 19 |
| UKR | 2017-08-01 | 26 | 89.1 | 2 | 239 | 800 | 15 | 1414 | 5.29 | 25.6 | 29.1 |
| UKR | 2017-08-23 | 74 | 65.6 | 2 | 215 | 700 | 12.8 | 889 | 5.31 | 21.8 | 21.1 |
| UKR | 2017-09-16 | 76 | 82 | 2 | 422 | 1000 | 20.6 | | 4.77 | 20.8 | 19.2 |
| UKR | 2018-05-10 | 37 | 78.1 | | 169 | 600 | 8.2 | 1414.2 | 5.31 | | 7.5 |
| UKR | 2018-06-07 | 59 | 73.3 | | 221 | 700 | 12.9 | 1414.2 | 5.34 | | 11.0 |
| UKR | 2018-07-05 | 99 | 66.3 | | 294 | 800 | 12.2 | 1414.2 | 5.46 | | 23.8 |
| UKR | 2018-10-02 | 47 | 77.3 | | 207 | 1100 | 10.5 | 1414.2 | 5.78 | | 7.7 |
| UKR | 2018-11-23 | 43 | 81.1 | | 227 | 800 | 10.8 | 1414.2 | 4.81 | | -9.1 |





Table A3 Linear correlation $r^2$ values and significance ($\alpha = 0.05$) between $Al_i/Al_d$ and other water chemistry parameters across all sites.

| Variable | Unit | Correlation with $Al_i/Al_d$ ($R^2$) | Significance (p-value) |
|---|---|---|---|
| $Al_d$ | $\mu g\ L^{-1}$ | 0.007 | 0.247 |
| Ca | $\mu g\ L^{-1}$ | 0.001 | 0.676 |
| DOC | $mg\ L^{-1}$ | 0.007 | 0.247 |
| pH | unit | 0.077 | 0.000 |
| Water Temp. | °C | 0.114 | 0.000 |
| $F^+$ | $\mu g\ L^{-1}$ | 0.003 | 0.537 |
| $NO_3^-$ | $\mu g\ L^{-1}$ | 0.002 | 0.624 |
| $SO_4^{2-}$ | $\mu g\ L^{-1}$ | 0.000 | 0.952 |





Table A4 Kendal-tau correlation and significance ($\alpha = 0.05$) between $Al_i$ and other water chemistry parameters for each study site. One $Al_i$ outlier removed for MR calculations (value: 2 µg L$^{-1}$, date: 30 April 2015).

| Site | Variable | Unit | Correlation Slope | Significance (p-value) |
|------|----------|------|-------------------|------------------------|
| ALD | Ald | µg L$^{-1}$ | 0.29 | 0.044 |
| | Ca | µg L$^{-1}$ | 0.22 | 0.143 |
| | DOC | mg L$^{-1}$ | 0.36 | 0.013 |
| | pH | unit | 0.19 | 0.190 |
| | Water Temp. | °C | 0.32 | 0.093 |
| | F$^+$ | µg L$^{-1}$ | 0.182 | 0.533 |
| | NO$_3^-$ | µg L$^{-1}$ | 0.600 | 0.142 |
| | SO$_4^{2-}$ | µg L$^{-1}$ | -0.037 | 0.876 |
| BLB | Ald | µg L$^{-1}$ | 0.03 | 0.852 |
| | Ca | µg L$^{-1}$ | 0.17 | 0.238 |
| | DOC | mg L$^{-1}$ | 0.08 | 0.575 |
| | pH | unit | 0.07 | 0.622 |
| | Water Temp. | °C | 0.35 | 0.099 |
| | F$^+$ | µg L$^{-1}$ | -0.036 | 0.901 |
| | NO$_3^-$ | µg L$^{-1}$ | -0.109 | 0.708 |
| | SO$_4^{2-}$ | µg L$^{-1}$ | -0.184 | 0.468 |
| CC | Ald | µg L$^{-1}$ | 0.11 | 0.708 |
| | Ca | µg L$^{-1}$ | -0.22 | 0.451 |
| | DOC | mg L$^{-1}$ | 0.25 | 0.383 |
| | pH | unit | -0.04 | 0.901 |
| | Water Temp. | °C | 0.67 | 0.174 |
| | F+ | µg L$^{-1}$ | | |
| | NO$_3^-$ | µg L$^{-1}$ | | |
| | SO$_4^{2-}$ | µg L$^{-1}$ | | |
| KB | Ald | µg L$^{-1}$ | 0.800 | 0.050 |
| | Ca | µg L$^{-1}$ | 0.200 | 0.624 |
| | DOC | mg L$^{-1}$ | 0.800 | 0.050 |
| | pH | unit | -0.200 | 0.624 |
| | Water Temp. | °C | 0.600 | 0.142 |
| | F+ | µg L$^{-1}$ | 0.800 | 0.050 |
| | NO$_3^-$ | µg L$^{-1}$ | | |





| | | | | |
|---|---|---|---|---|
| | SO$_4^{2-}$ | µg L$^{-1}$ | -0.400 | 0.327 |
| LR | Ald | µg L$^{-1}$ | 0.37 | 0.047 |
| | Ca | µg L$^{-1}$ | 0.24 | 0.226 |
| | DOC | mg L$^{-1}$ | 0.25 | 0.189 |
| | pH | unit | 0.19 | 0.319 |
| | Water Temp. | °C | 0.02 | 0.937 |
| | F+ | µg L$^{-1}$ | | |
| | NO$_3^-$ | µg L$^{-1}$ | -0.333 | 0.348 |
| | SO$_4^{2-}$ | µg L$^{-1}$ | 0.105 | 0.801 |
| MB | Ald | µg L$^{-1}$ | 0.739 | 0.001 |
| | Ca | µg L$^{-1}$ | -0.062 | 0.783 |
| | DOC | mg L$^{-1}$ | 0.400 | 0.073 |
| | pH | unit | -0.279 | 0.214 |
| | Water Temp. | °C | 0.125 | 0.580 |
| | F+ | µg L$^{-1}$ | -0.028 | 0.917 |
| | NO$_3^-$ | µg L$^{-1}$ | -0.182 | 0.533 |
| | SO$_4^{2-}$ | µg L$^{-1}$ | -0.463 | 0.050 |
| MPB | Ald | µg L$^{-1}$ | 0.550 | 0.000 |
| | Ca | µg L$^{-1}$ | 0.580 | 0.000 |
| | DOC | mg L$^{-1}$ | 0.574 | 0.000 |
| | pH | unit | -0.169 | 0.146 |
| | Water Temp. | °C | 0.280 | 0.016 |
| | Runoff | mm day$^{-1}$ | -0.232 | 0.042 |
| | F+ | µg L$^{-1}$ | 0.239 | 0.042 |
| | NO$_3^-$ | µg L$^{-1}$ | 0.190 | 0.160 |
| | SO$_4^{2-}$ | µg L$^{-1}$ | -0.206 | 0.067 |
| MR | Ald | µg L$^{-1}$ | 0.459 | 0.000 |
| | Ca | µg L$^{-1}$ | 0.317 | 0.002 |
| | DOC | mg L$^{-1}$ | 0.382 | 0.000 |
| | pH | unit | 0.097 | 0.362 |
| | Water Temp. | °C | 0.285 | 0.007 |
| | RunOff | mm day$^{-1}$ | -0.108 | 0.291 |
| | F+ | µg L$^{-1}$ | 0.139 | 0.188 |
| | NO$_3^-$ | µg L$^{-1}$ | 0.086 | 0.450 |
| | SO$_4^{2-}$ | µg L$^{-1}$ | -0.127 | 0.215 |
| PMB | Ald | µg L$^{-1}$ | 0.46 | 0.019 |





| | | | | |
|---|---|---|---|---|
| | Ca | µg L$^{-1}$ | 0.01 | 0.960 |
| | DOC | mg L$^{-1}$ | 0.21 | 0.295 |
| | pH | unit | -0.23 | 0.232 |
| | Water Temp. | °C | 0.36 | 0.065 |
| | F+ | µg L$^{-1}$ | -0.063 | 0.782 |
| | NO$_3^-$ | µg L$^{-1}$ | 0.276 | 0.444 |
| | SO$_4^{2-}$ | µg L$^{-1}$ | -0.293 | 0.135 |
| | Ald | µg L$^{-1}$ | 0.34 | 0.071 |
| | Ca | µg L$^{-1}$ | 0.38 | 0.053 |
| | DOC | mg L$^{-1}$ | 0.32 | 0.086 |
| UKR | pH | unit | 0.35 | 0.063 |
| | Water Temp. | °C | 0.14 | 0.621 |
| | F+ | µg L$^{-1}$ | | |
| | NO$_3^-$ | µg L$^{-1}$ | | |
| | SO$_4^{2-}$ | µg L$^{-1}$ | -0.600 | 0.142 |





Table A5 $R^2$ values for scatterplots of water chemistry relationships shown in Figure 3

| Site | Season | Season Dates | Relationship | $R^2$ |
|------|--------|--------------|--------------|-------|
| MR | S1 | April-May | $Al_i$-pH | 0.78131 |
| MR | S2 | June-Aug | $Al_i$-pH | 0.27845 |
| MR | S3 | Sept-Feb | $Al_i$-pH | 0.04551 |
| MR | S1 | April-May | $Al_i$-DOC | 0.48879 |
| MR | S2 | June-Aug | $Al_i$-DOC | 0.51343 |
| MR | S3 | Sept-Feb | $Al_i$-DOC | 0.0014 |
| MR | S1 | April-May | $Al_i$-$T_w$ | 0.42004 |
| MR | S2 | June-Aug | $Al_i$-$T_w$ | 0.03442 |
| MR | S3 | Sept-Feb | $Al_i$-$T_w$ | 0.08795 |
| MR | S1 | April-May | $Al_i$-$Al_d$ | 0.66782 |
| MR | S2 | June-Aug | $Al_i$-$Al_d$ | 0.52313 |
| MR | S3 | Sept-Feb | $Al_i$-$Al_d$ | 0.0141 |
| MR | S1 | April-May | $Al_i$-Ca | 0.50399 |
| MR | S2 | June-Aug | $Al_i$-Ca | 0.37339 |
| MR | S3 | Sept-Feb | $Al_i$-Ca | 0.00009 |
| MR | S1 | April-May | $Al_i$-Ca/$Al_d$ | 0.41377 |
| MR | S2 | June-Aug | $Al_i$-Ca/$Al_d$ | 0.32486 |
| MR | S3 | Sept-Feb | $Al_i$-Ca/$Al_d$ | 0.0382 |
| MR | S1 | April-May | $Al_i$-Q | 0.0374 |
| MR | S2 | June-Aug | $Al_i$-Q | 0.0703 |
| MR | S3 | Sept-Feb | $Al_i$-Q | 0.0063 |





| MR | S1 | April-May | $Al_d$-Ca | 0.55308 |
| MR | S2 | June-Aug | $Al_d$-Ca | 0.63892 |
| MR | S3 | Sept-Feb | $Al_d$-Ca | 0.5074 |
| MPB | S1 | April-June | $Al_i$-pH | 0.00447 |
| MPB | S2 | July-Aug | $Al_i$-pH | 0.21629 |
| MPB | S3 | Sept-Oct | $Al_i$-pH | 0.56 |
| MPB | S1 | April-June | $Al_i$-DOC | 0.70785 |
| MPB | S2 | July-Aug | $Al_i$-DOC | 0.43036 |
| MPB | S3 | Sept-Oct | $Al_i$-DOC | 0.72722 |
| MPB | S1 | April-June | $Al_i$-$T_w$ | 0.72067 |
| MPB | S2 | July-Aug | $Al_i$-$T_w$ | 0.2356 |
| MPB | S3 | Sept-Oct | $Al_i$-$T_w$ | 0.4353 |
| MPB | S1 | April-June | $Al_i$-$Al_d$ | 0.67571 |
| MPB | S2 | July-Aug | $Al_i$-$Al_d$ | 0.4225 |
| MPB | S3 | Sept-Oct | $Al_i$-$Al_d$ | 0.65683 |
| MPB | S1 | April-June | $Al_i$-Ca | 0.59175 |
| MPB | S2 | July-Aug | $Al_i$-Ca | 0.4214 |
| MPB | S3 | Sept-Oct | $Al_i$-Ca | 0.49111 |
| MPB | S1 | April-June | $Al_i$-Ca/$Al_d$ | 0.51142 |
| MPB | S2 | July-Aug | $Al_i$-Ca/$Al_d$ | 0.03067 |
| MPB | S3 | Sept-Oct | $Al_i$-Ca/$Al_d$ | 0.02961 |
| MPB | S1 | April-June | $Al_i$-Q | 0.1734 |
| MPB | S2 | July-Aug | $Al_i$-Q | 0.0039 |
| MPB | S3 | Sept-Oct | $Al_i$-Q | 0.0004 |
| MPB | S1 | April-June | $Al_d$-Ca | 0.96289 |
| MPB | S2 | July-Aug | $Al_d$-Ca | 0.7685 |



| MPB | S3 | Sept-Oct | $Al_d$-Ca | 0.72173 |
|-----|----|----|----|----|





Table A6  Laboratory detection limit comparison.

| Chemistry Parameter | Units | Value | | |
| --- | --- | --- | --- | --- |
| | | HERC | Maxxam | AGAT |
| pH | $\mu g\ L^{-1}$ | n/a | n/a | n/a |
| DOC | mg L$^{-1}$ | n/a | 0.50 | n/a |
| TOC | mg L$^{-1}$ | n/a | n/a | 0.5 |
| SO$_4$ | $\mu g\ L^{-1}$ | 10.00 | n/a | 2000 |
| Al$_d$ | $\mu g\ L^{-1}$ | n/a | 5.00 | 5 |
| Al$_t$ | $\mu g\ L^{-1}$ | n/a | 5.00 | 5 |
| Al$_o$ | $\mu g\ L^{-1}$ | n/a | 5.00 | 5 |
| Ca$_t$ | $\mu g\ L^{-1}$ | n/a | 100 $\mu g\ L^{-1}$ | 0.1 mg L$^{-1}$ |
| Ca$_d$ | $\mu g\ L^{-1}$ | n/a | 100 | 100 |



# Appendix B.  Figures

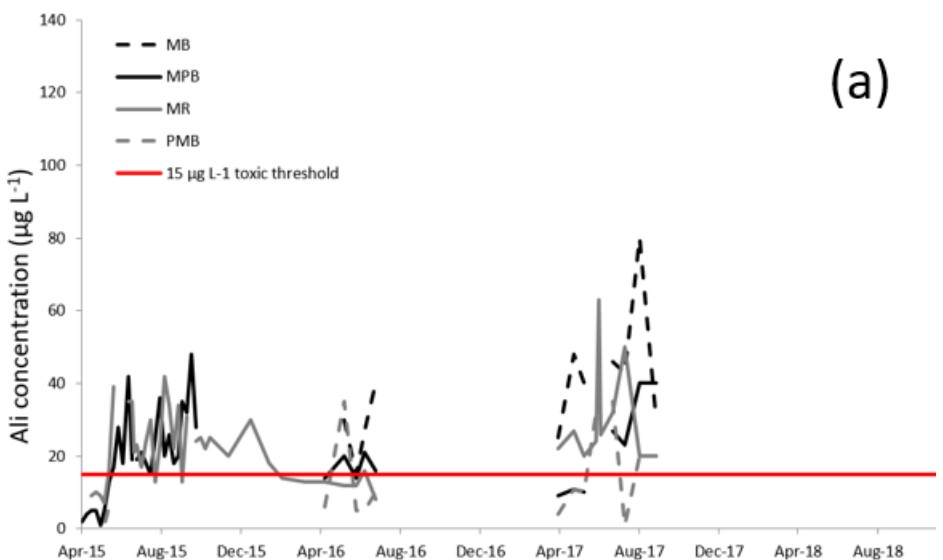

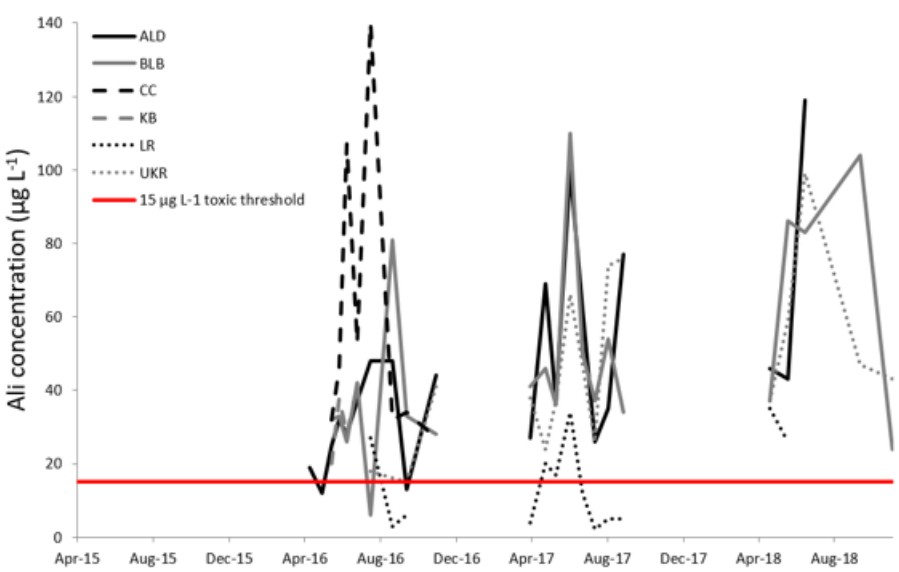

Figure B1 Timeseries of Al$_i$ concentration between 22 April 2015 and 23 November 2018.



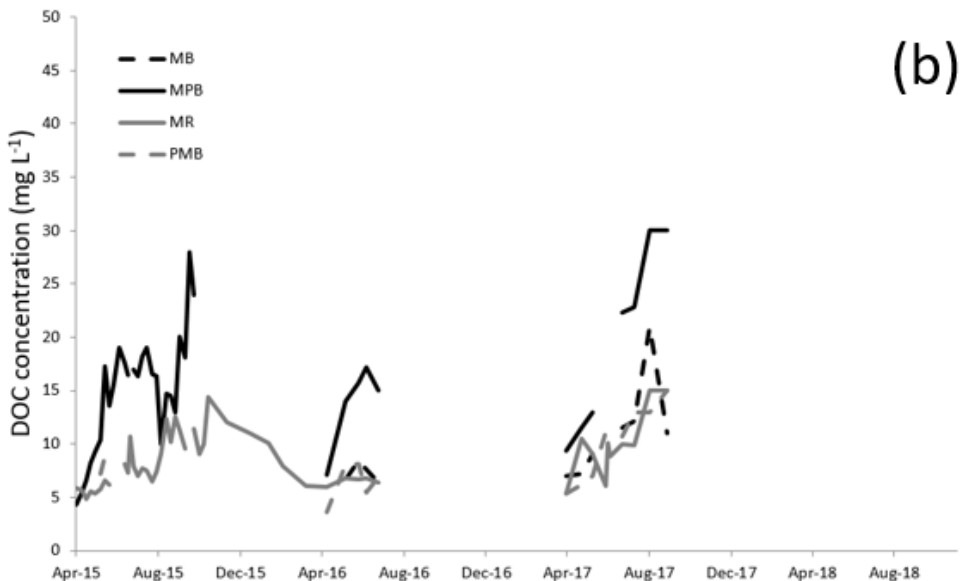

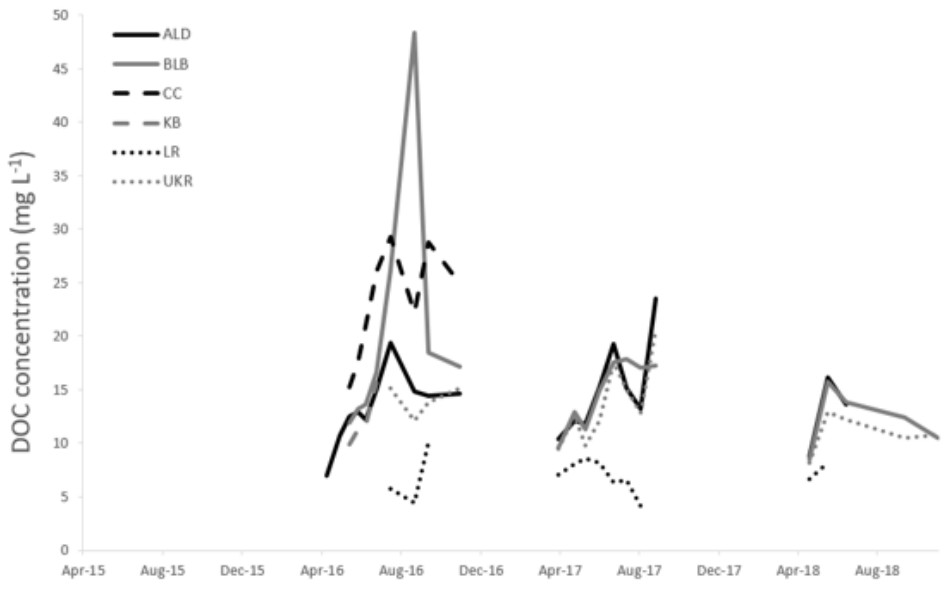

Figure B2 Time series of DOC concentration between 22 April 2015 and 23 November 2018





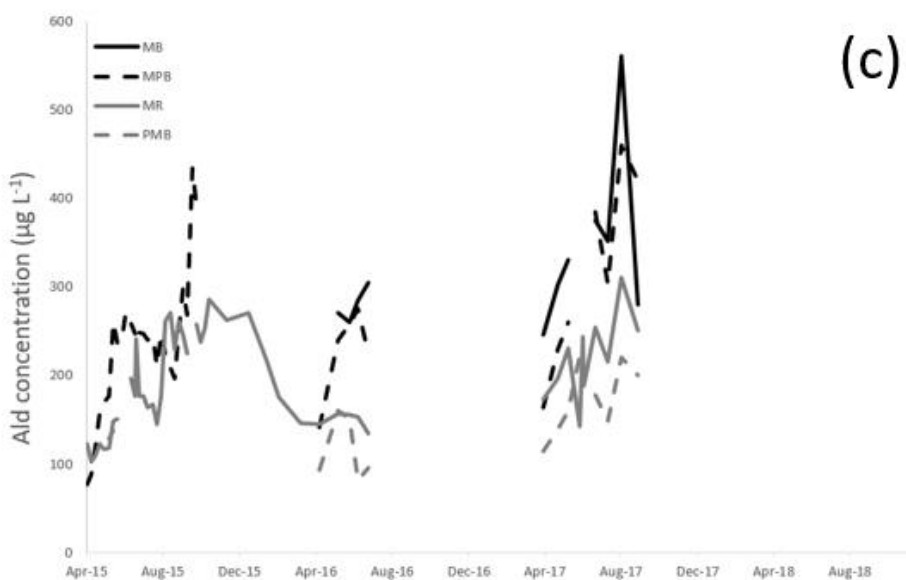

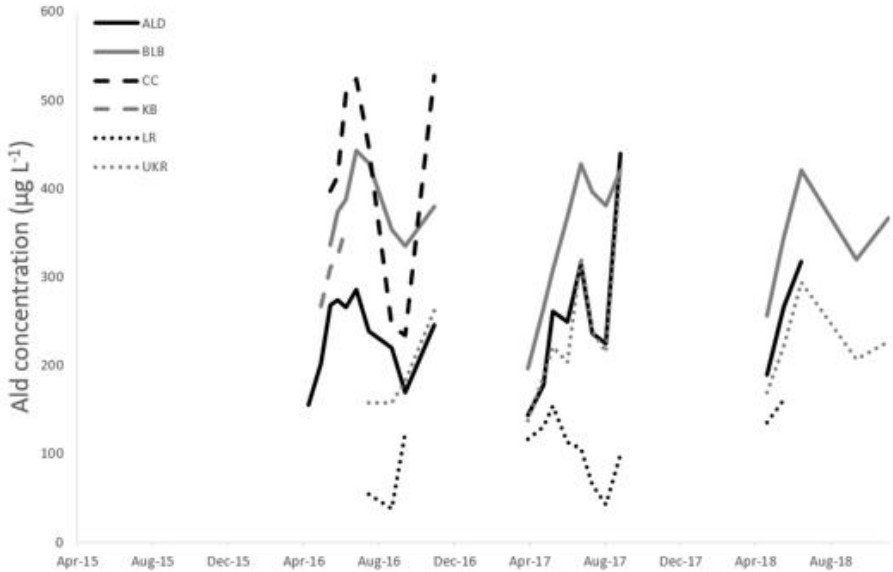

Figure B3 Time series of $Al_d$ concentration between 22 April 2015 and 23 November 2018.





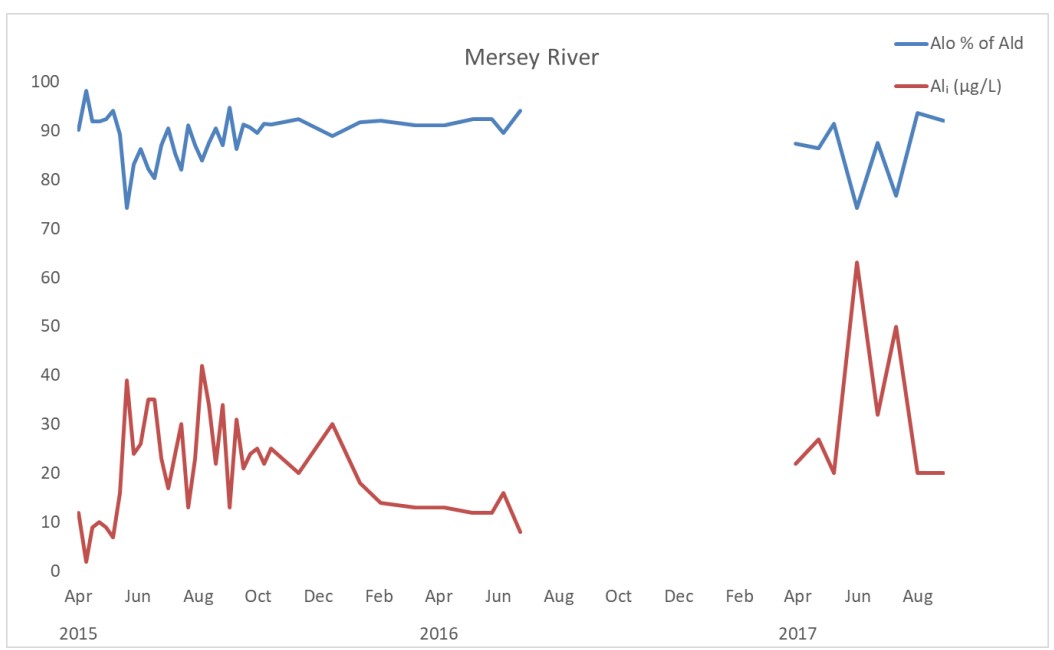

Figure B4 Time series of percentage $Al_d$ comprised of $Al_o$ for MR, compared to absolute value of $Al_i$ in ug $L^{-1}$.

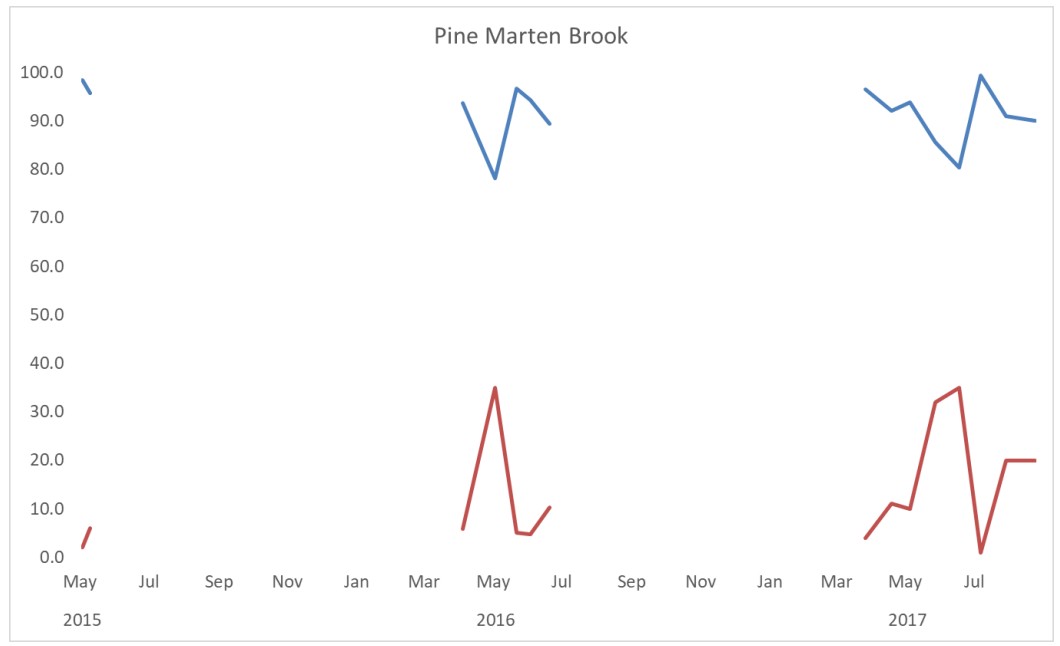





Figure B5 Time series of percentage $Al_d$ comprised of $Al_o$ for PMB, compared to absolute value of $Al_i$ in ug L$^{-1}$.

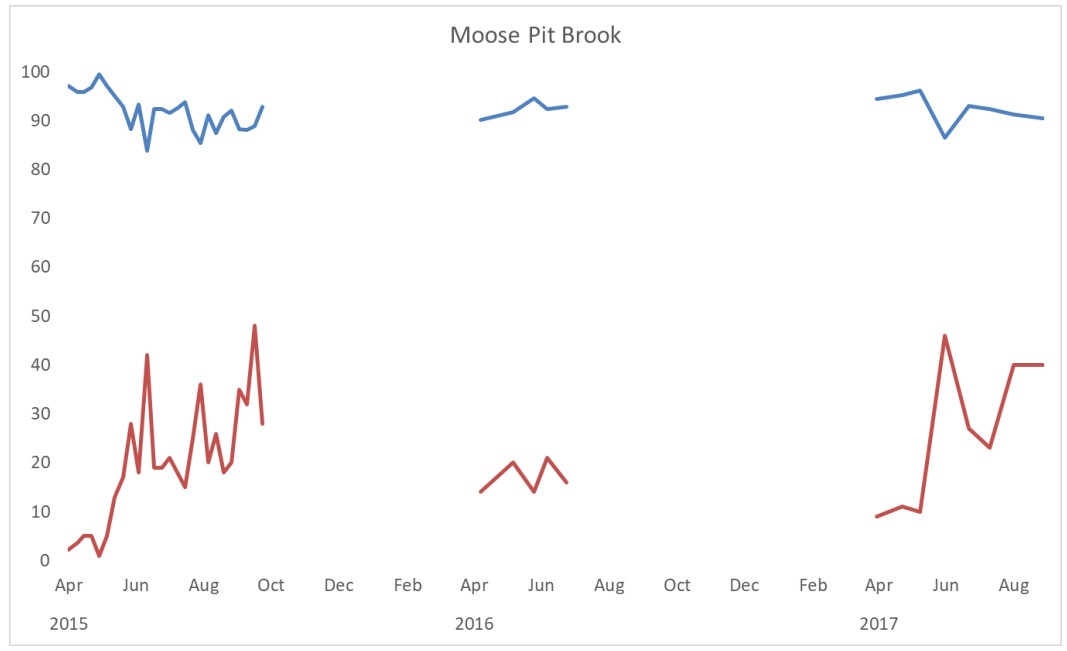

Figure B6 Time series of percentage $Al_d$ comprised of $Al_o$ for MPB, compared to absolute value of $Al_i$ in ug L$^{-1}$.





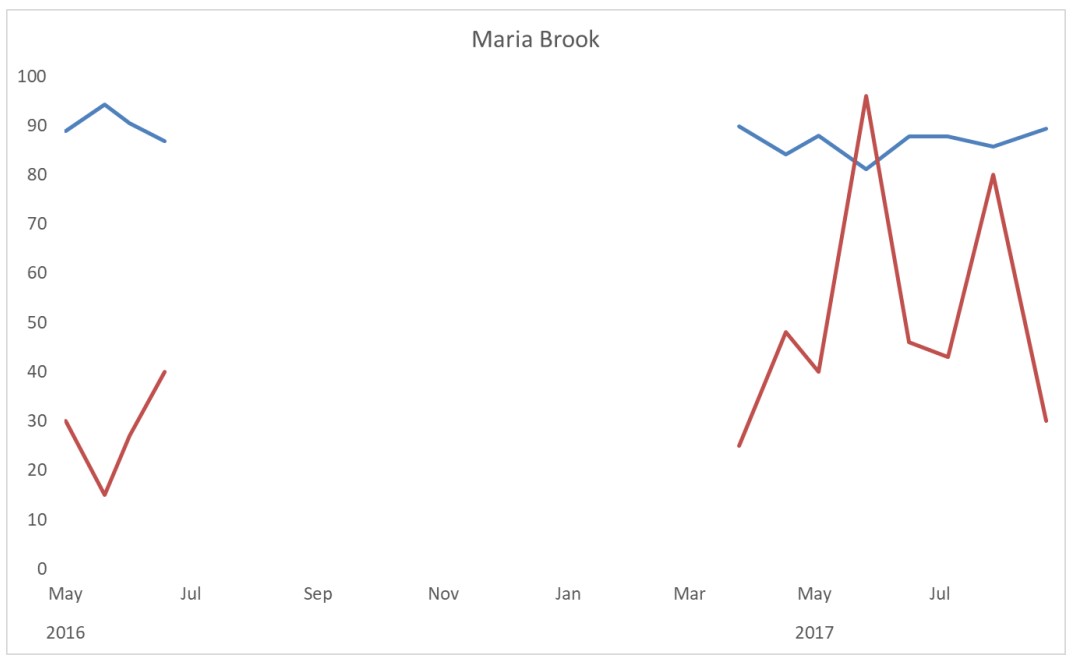

Figure B7 Time series of percentage $Al_d$ comprised of $Al_o$ for MB, compared to absolute value of $Al_i$ in ug $L^{-1}$.

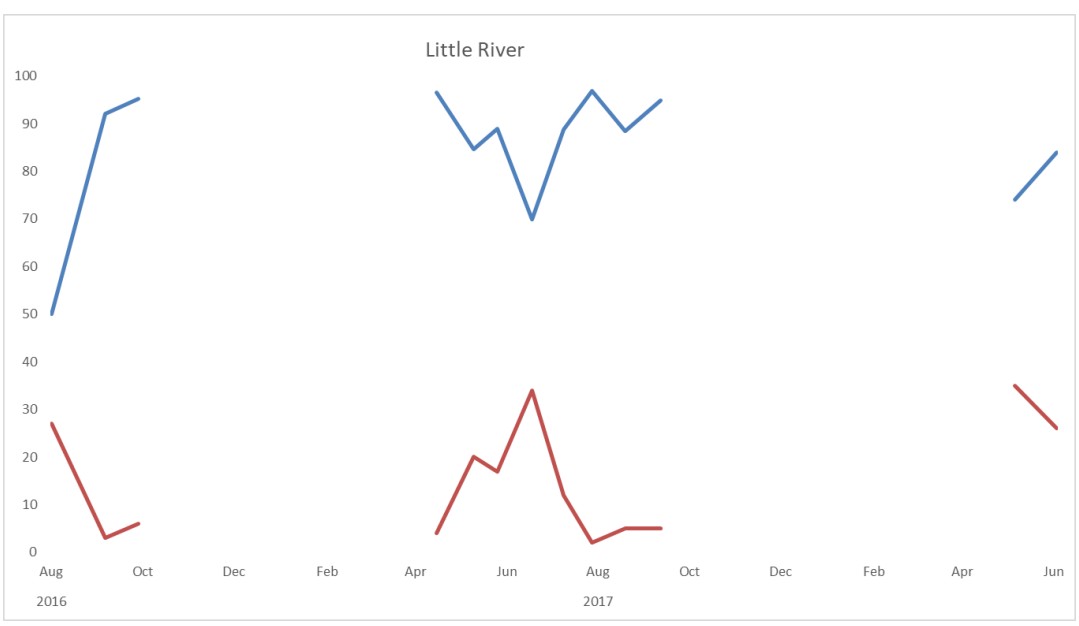





Figure B8 Time series of percentage $Al_d$ comprised of $Al_o$ for LR, compared to absolute value of $Al_i$ in ug $L^{-1}$.

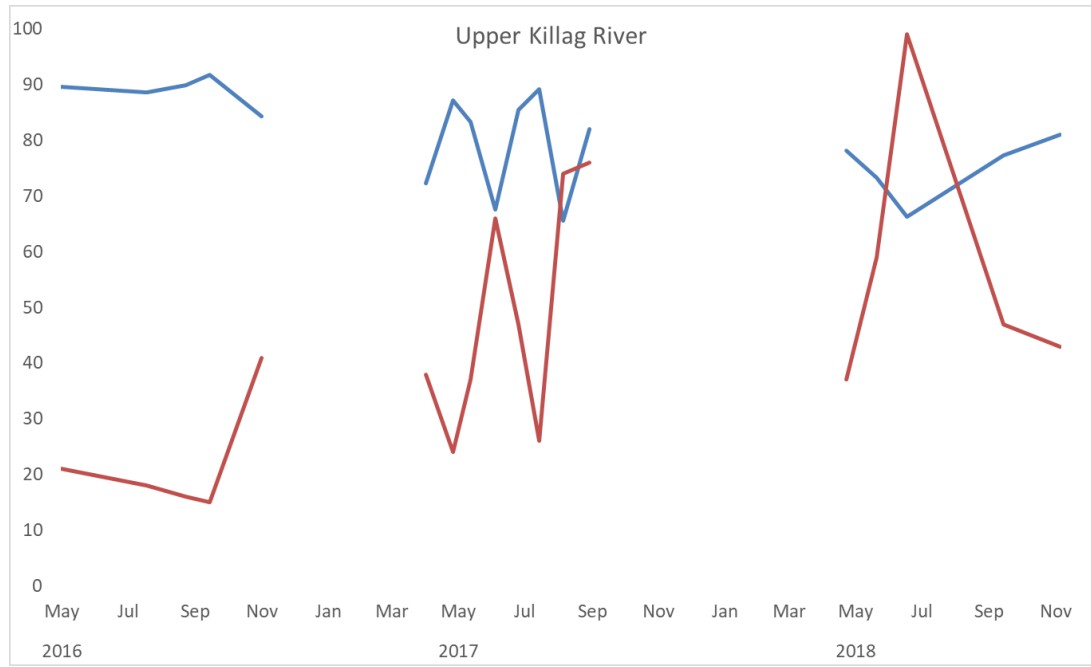

Figure B9 Time series of percentage $Al_d$ comprised of $Al_o$ for UKR, compared to absolute value of $Al_i$ in ug $L^{-1}$.



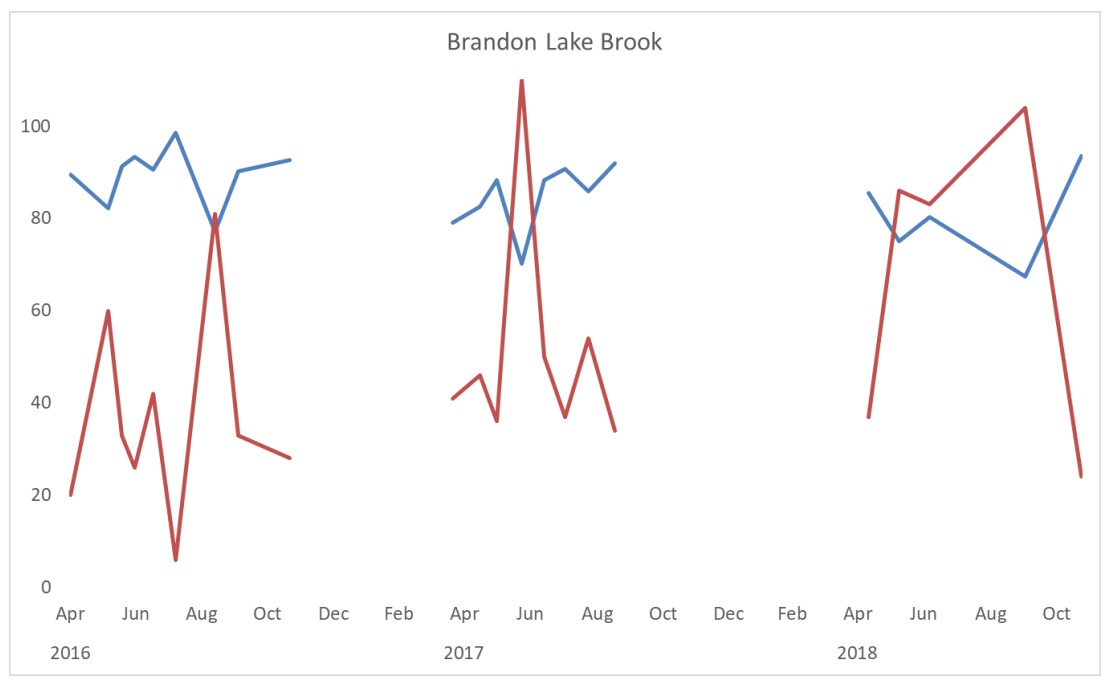

Figure B10 Time series of percentage $Al_d$ comprised of $Al_o$ for BLB, compared to absolute value of $Al_i$ in ug $L^{-1}$.

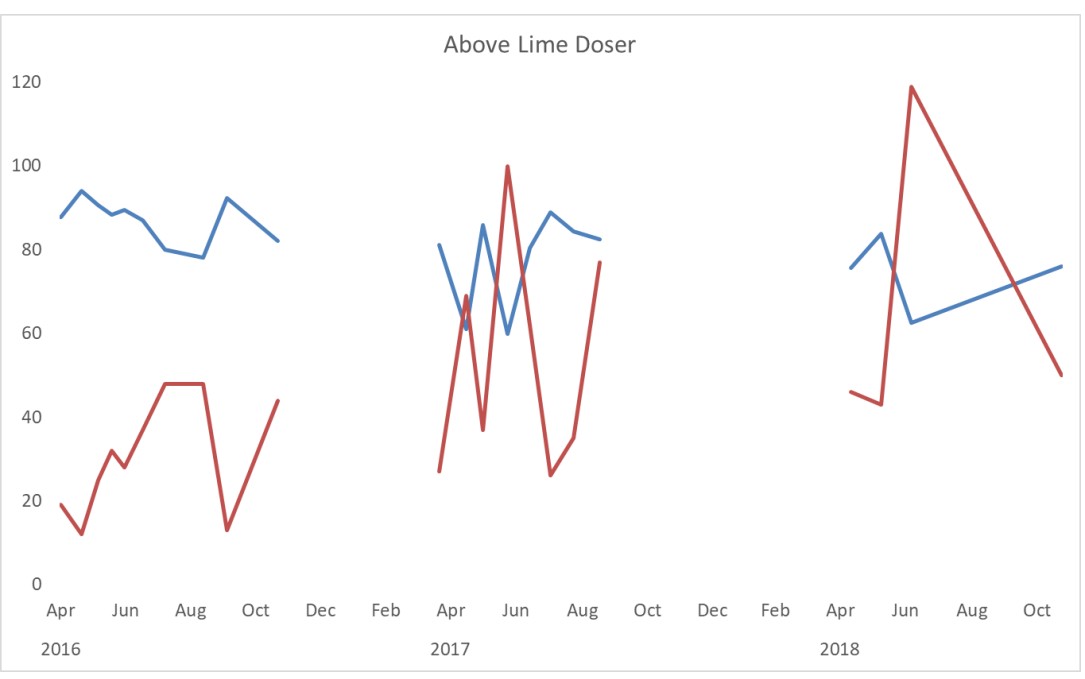



Figure B11 Time series of percentage $Al_d$ comprised of $Al_o$ for ALD, compared to absolute value of $Al_i$ in ug $L^{-1}$.

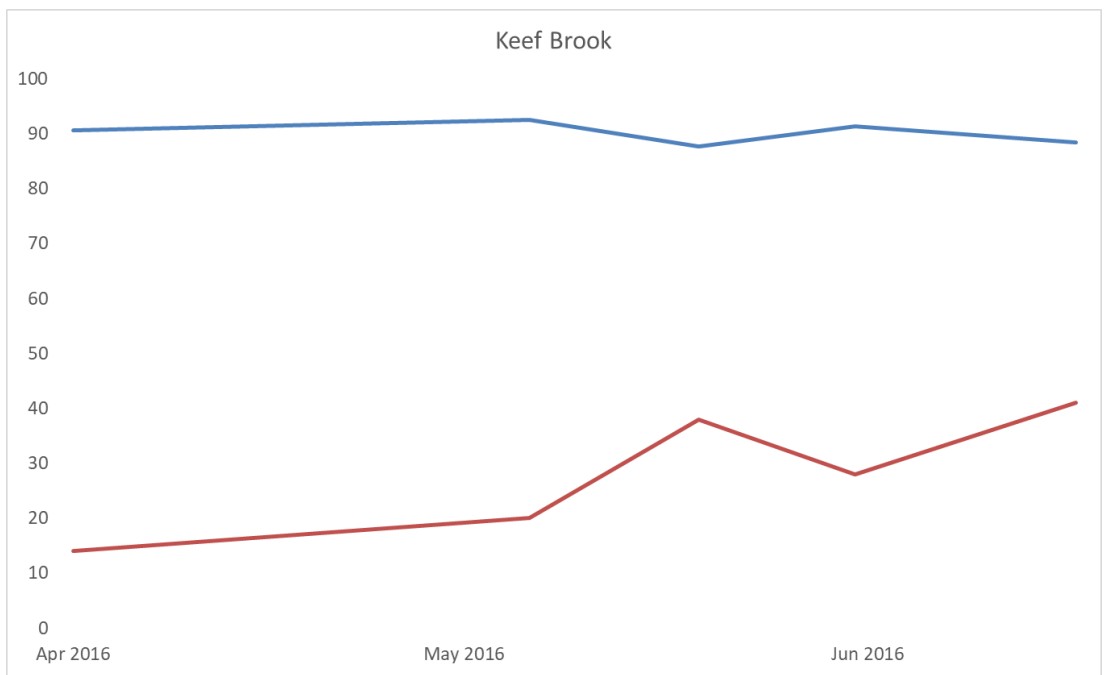

Figure B12 Time series of percentage $Al_d$ comprised of $Al_o$ for KB, compared to absolute value of $Al_i$ in ug $L^{-1}$.





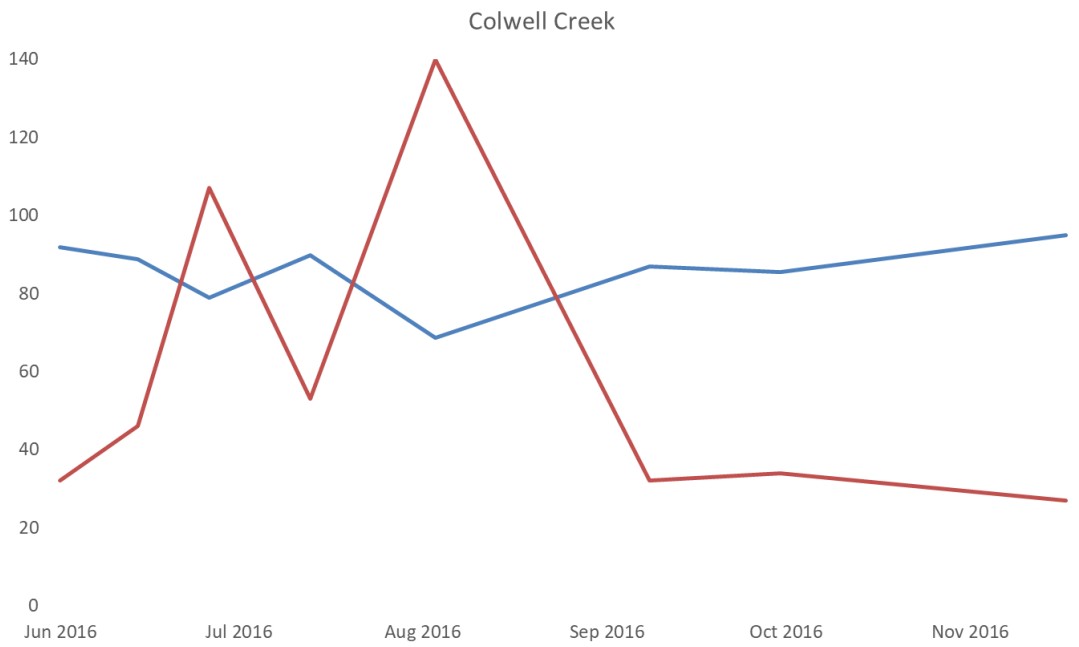

Figure B13 Time series of percentage $Al_d$ comprised of $Al_o$ for CC, compared to absolute value of $Al_i$ in ug $L^{-1}$

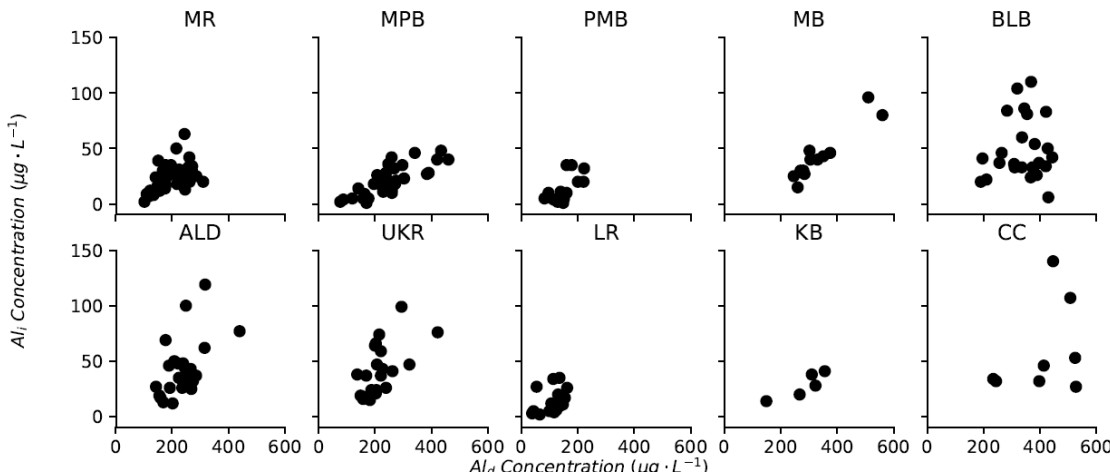

Figure B14 Least-squares linear regression of $Al_i$ versus $Al_d$ for each study site. One $Al_i$ outlier removed for MR

(value: 2 µg L-1, date: 30 April 2015).





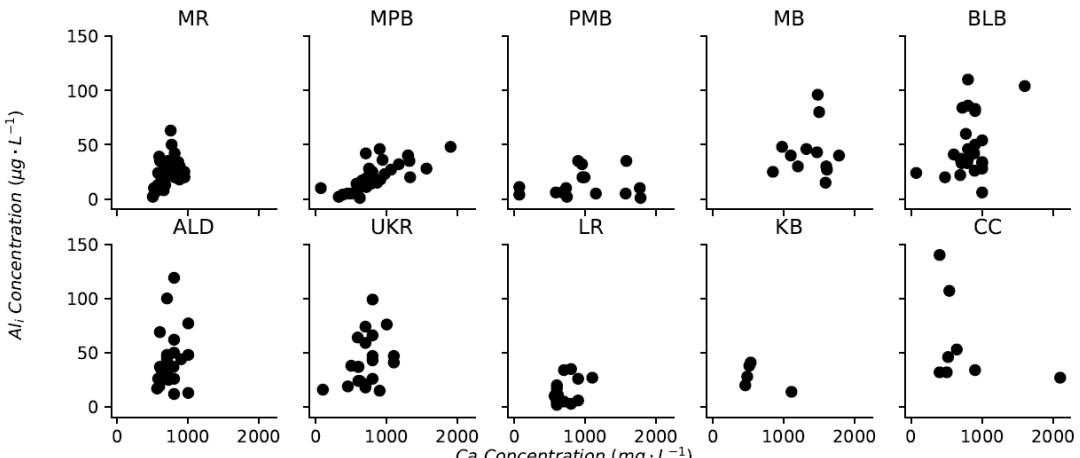

Figure B15 Least-squares linear regression of Al$_i$ versus Ca for each study site. One Al$_i$ outlier removed for MR

(value: 2 µg L-1, date: 30 April 2015). One Ca outlier for KB removed (value: 1110 µg L-1, date: 29 April 2016).

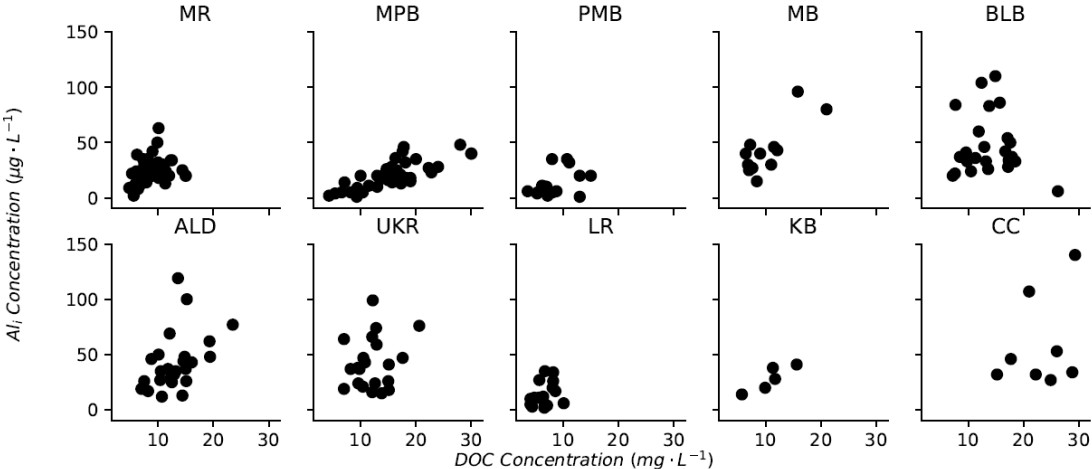

Figure B16 Least-squares linear regression of Al$_i$ versus DOC for each study site. One Al$_i$ outlier removed for MR

(value: 2 µg L-1, date: 30 April 2015).





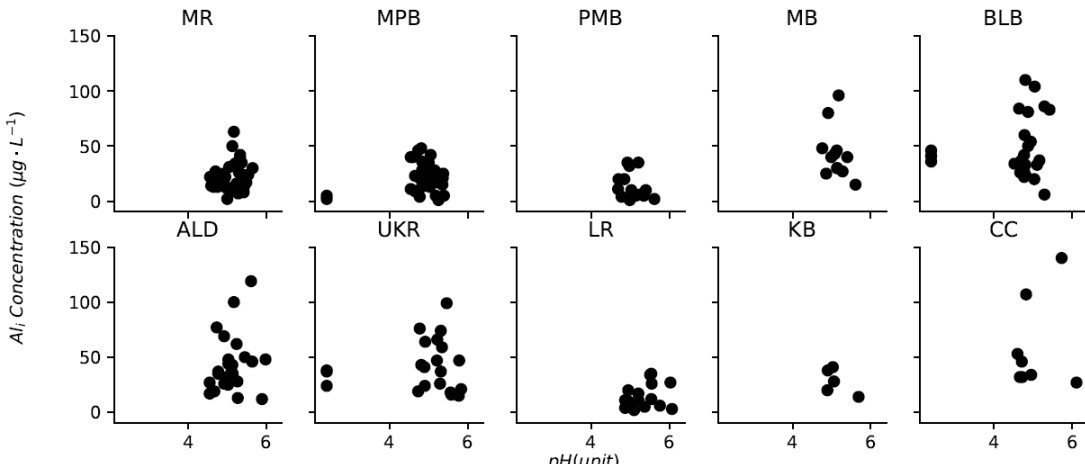

Figure B17 Least-squares linear regression of Al$_i$ versus pH for each study site. One Al$_i$ outlier removed for MR

(value: 2 µg L-1, date: 30 April 2015).

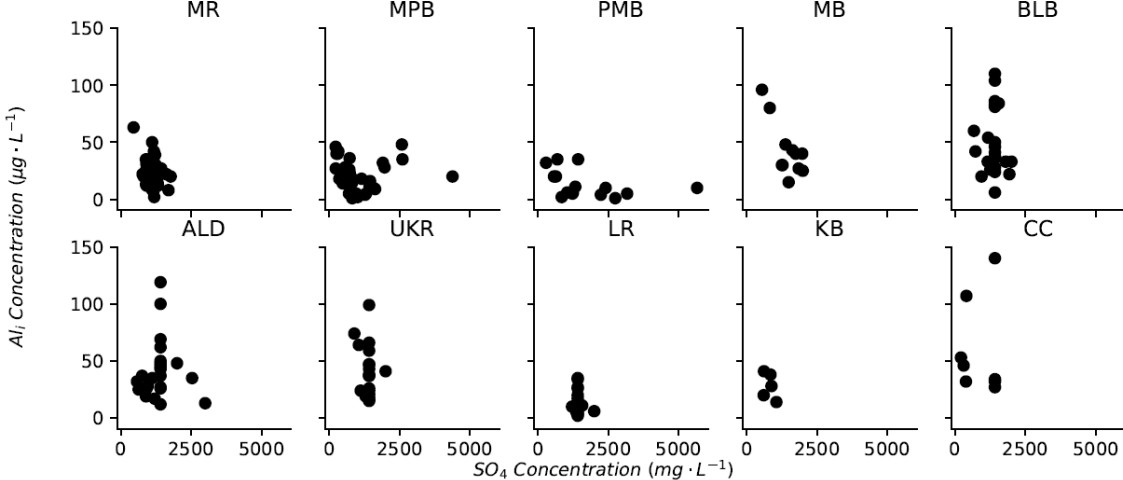

Figure B18 Least-squares linear regression of Al$_i$ versus SO$_4^{2-}$ for each study site. One Al$_i$ outlier removed for MR

(value: 2 µg L-1, date: 30 April 2015).





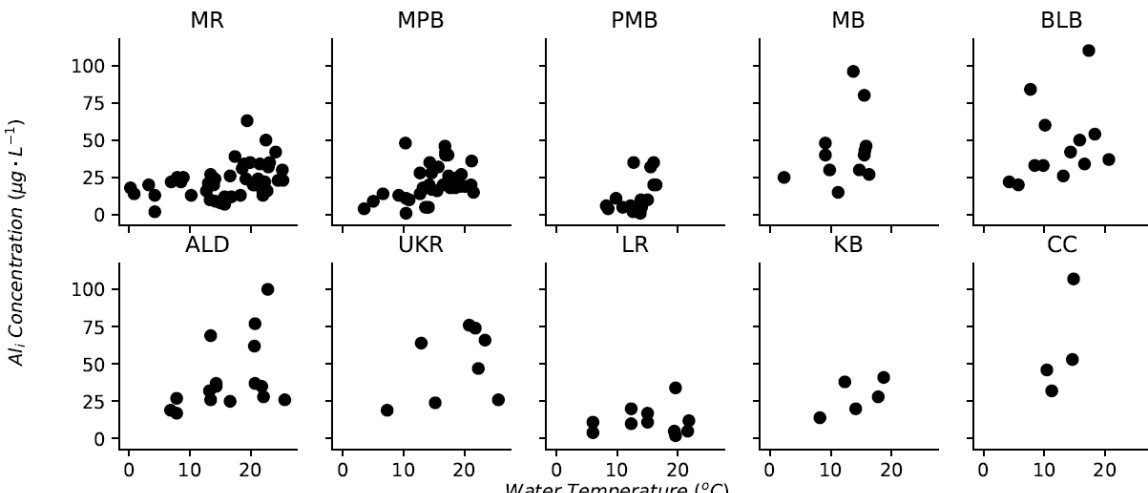

Figure B19 Least-squares linear regression of Al$_i$ versus T$_w$ for each study site. One Al$_i$ outlier removed for MR

(value: 2 µg L-1, date: 30 April 2015).

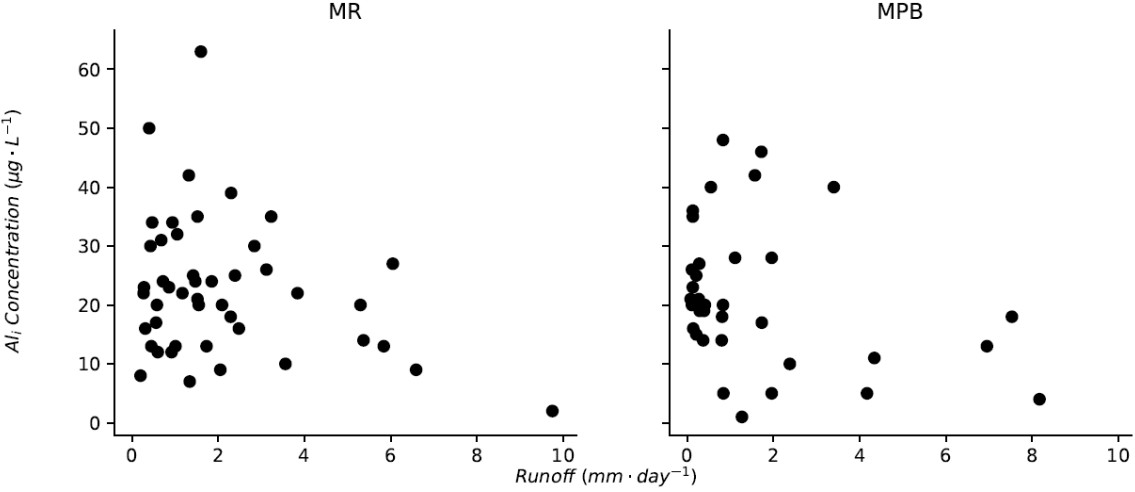

Figure B20 Least-squares linear regression of Al$_i$ versus runoff for each study site. One Al$_i$ outlier removed for MR

(value: 2 µg L-1, date: 30 April 2015). One runoff outlier for MR removed (value: 17.294 mm day-1, date: 22 April

2015), and one runoff outlier for MPB removed (value: 34.994 mm day-1, date: 22 April 2015).





# Appendix C. Scripts

C.1. Linear regression
```
"""Linear regression calculation script
:author: Lobke Rotteveel
:email: lobke.rotteveel@dal.ca
"""

# Import modules
from scipy import stats
import pandas as pd
import csv

# Import data
df = pd.read_csv('Input.csv')

# Run Mann Kendall test on site-variable groups and create table of results
results = []
results.append(['site_id', 'variable', 'tau', 'pvalue', 'slope', 'std error of slope'])
grouped = df.groupby('Site')
for name, group in grouped:
    chem_groups = [group['Ald'], group['Ca'], group['DOC_TOC'], group['CalibpH'],
group['Tw'], group['RunOff']]

    Ali = group['Ali']
    for i in chem_groups:
        pair = {'i':i,'Ali':Ali}
        pair = pd.DataFrame(pair)
        pair = pair.dropna()
        if not pair.empty:
            ken_tau = stats.kendalltau(pair['i'], pair['Ali'])
            slope = stats.linregress(pair['i'], pair['Ali'])
            result_row = [name, i.name, ken_tau.correlation, ken_tau.pvalue, slope.slope,
slope.stderr]
            results.append(result_row)

with open('LinearRegression_Out.csv', 'w') as f:
    writer = csv.writer(f)
    writer.writerows(results)
```

C.2. Laboratory comparison
```
"""Laboratory result comparison script
:author: Lobke Rotteveel
:email: lobke.rotteveel@dal.ca
```





```
"""

# Import modules
import pandas as pd
import numpy as np
import scipy as sp
from scipy import stats
import warnings

warnings.simplefilter('ignore', np.RankWarning)

# Importing data
df = pd.read_csv('SampDat_CompareInput_LimSur_171105_LR.csv', ',', header=0)
#print (df.head(n=5))

# Run comparisson
with open('SampData_Compare_LimSur.txt', 'w') as f:

    x = df.filter(regex='B_.*').columns
    y = df.filter(regex='A_.*').columns

    for x_col, y_col in zip(x,y):
        Sig = sp.stats.wilcoxon(df[x_col],df[y_col])
        f.write('x: {}, y: {}, sig:{}\n'.format(x_col, y_col, Sig))
```



# Appendix D. Additional methods

D.1 Laboratory analysis methods

Samples were analyzed at Maxxam Analytics Laboratory, Health and Environmental
Research Centre (HERC), and AGAT Laboratories. Samples from MR, MPB, PMB, MB, KB,
and CC were analyzed at Maxxam and HERC labs only. Samples from BLB, ALD, UKR, and
LR were analyzed at all three labs.

D1.1 Maxxam Laboratory

The protocol at Maxxam Laboratory in Bedford, NS, adheres to methods approved by the
United States Environmental Protection Agency (US EPA) for identifying trace elements in
water (US EPA, 1994) and analyzing samples using Inductively Coupled Plasma-Mass
Spectrometry (ICP-MS) (US EPA, 1998). Cations and anions were analyzed using ICP-MS,
while a Continuous Flow Analyzer was used to measure DOC. pH was measured using a
standard hydrogen electrode and reference electrode.

D1.2 HERC Laboratory

$SO_4^{2-}$ samples were analyzed at HERC Laboratory in Halifax, NS, due to lower detection
limits at the Maxxam laboratory. Once delivered to the laboratory, samples were filtered using a
0.45 µm glass fiber filter and analyzed using an Ion-Chromatography System (ICS) 5000 Dionex
detector.

D1.3 AGAT Laboratory

Samples collected in the West River, Sheet Harbour area (UKR, ALD, LR, BLB, KB,
CC) were analyzed at the AGAT laboratory in Dartmouth, NS. This laboratory holds the





9001:2015 and 17025:2005 International Organization for Standardization accreditations. Cation samples were analyzed using ICP-MS, laboratory pH was measured using a standard hydrogen electrode and reference electrode, and $SO_4^{2-}$ and anions were measured using ICS. Samples analyzed at AGAT were analyzed for total organic carbon (TOC) as opposed to DOC and were analyzed using Infrared Combustion (IR Combustion).

D.2 Data quality assurance and control

Blanks were used to assess contamination during the $Al_o$ extraction procedure. Blanks were collected on 10% of samples, taken on arbitrary sampling events. Triple deionized water was collected before passing through filter and column ("Blank Before"), and after ("Blank After"). The triple-deionized water had traces of chemicals below the laboratory detection limits, providing "Not Detectable" results for the Blank Before sample. If chemicals were detected in the Blank After sample, this would have indicated leaching of chemicals from the column.

Duplicates were collected and analyzed for 10% of the samples; on arbitrarily selected sampling events, $Al_o$ and $Al_{filtered}$ or $Al_{unfiltered}$, were analyzed twice, independently, by Maxxam laboratory. All laboratories also conducted additional duplicate, blank, reference material, and matrix spike testing, in addition to instrument calibration in adherence to industry standards for quality control and assurance.

To verify that sample analysis results from the Maxxam/HERC laboratory combination were comparable to AGAT, three sets of duplicate samples were collected for ALD, BLB, UKR, and LR (19 April 2017, 14 May 2017, and 30 May 2017) and analyzed by both laboratories. Laboratory results were compared using Wilcoxon Rank Sum statistical test in Python 3.6.5 using the SciPy Stats module (version 0.19) (Appendix C.2). Results indicated a significant difference in pH values between laboratories (T = 1, p = 0.04), therefore, statistical analysis on



pH data was conducted on the calibrated YSI Pro Plus sonde field data. $Al_o$, $Al_{filtered}$, and $Al_{unfiltered}$ results were found to be comparable between laboratories (T = 8.5, p = 0.674; T = 5.0, p = 0.249; and T = 8.0, p = 0.600, respectively). After adjusting for detection limits (Table A6), Ca results were also found to be comparable between laboratories (T = 4.0, p = 0.173). However, due to the large difference in $SO_4^{2-}$ detection limits between HERC and AGAT (10 µg L$^{-1}$ and 2 mg L$^{-1}$, respectively), results for $SO_4^{2-}$ are not comparable between laboratories. Lastly, organic carbon analyzed at Maxxam was analyzed for DOC, while AGAT analyzed for TOC, therefore these results cannot be compared. For dates where duplicate data is present, AGAT data was used to maintain data source consistency, apart from $SO_4^{2-}$ data, for which HERC data was used due to superior detection limits. Analysis for BLB and ALD transitioned from Maxxam to AGAT 19 April 2017 and consequently DOC is approximated as TOC for these two sites after this date.

The YSI Pro Plus sonde was calibrated within 36 hours of in-stream data collection.

D.3 Toxic thresholds of $Al_i$

Identified toxic thresholds of $Al_i$ for *Salmo salar* vary in the literature. Based on toxicological and geochemical studies on Al and *Salmo salar*, the EIFAC suggested an $Al_i$ toxic threshold of 15 ug L$^{-1}$ for Atlantic salmon in freshwaters for pH between 5.0 and 6.0, and 30 ug L$^{-1}$ in pH <5 (Howells et al., 1990). The lower threshold at higher pH is to account for the increased fraction in the $Al(OH)_2^+$ species. At pH > 6, the toxic effects of $Al_i$ to *Salmo salar* are considered negligible, and toxic effects are dominated by other dissolved and precipitated forms (Gensemer et al., 2018), due to the decreased solubility of Al at pH > 6 (Dennis and Clair 2012). However, in colder rivers, the pH-toxicity threshold may be higher, closer to pH 6.5 (Lydersen,

1990). For the purposes of this study, we use the toxic threshold of $Al_i$ at 15 ug $L^{-1}$, as the

majority of our pH observations were greater than or equal to 5.0 (Table A2).

D.4 Calibration of pH measurements

In situ pH measurements were taken using a YSI Pro Plus sonde and confirmed with a

YSI Ecosense pH Pen. It was found that measurements taken with the YSI Pro Plus sonde

deviated from the YSI Ecosense Pen, which is known to measure pH accurately ($0.47 \pm 0.44$ pH

units below in-stream pH as measured by YSI Ecosense Pen). Therefore, a calibration curve was

created based on simultaneous side-by-side measurements of both instruments (n = 69 pairs) and

the in situ pH data were adjusted accordingly (Eq. 1).

$$YSI\ Ecosense\ Pen\ pH = 0.595(Pro\ Plus\ pH) + 2.3868 \qquad (1)$$