# Peer review of "Ionic aluminium concentrations exceed thresholds for"

_Hydrology and Earth System Sciences, 2019_

## Referee Comment (RC1) · Anonymous Referee #1 · 20 Nov 2019

Overall this is a well written manuscript that contributes new information on an important topic in relation to aquatic ecosystem risks from labile Al.

Line 134 - re: Bond Elut Jr column - I am unfamiliar with these specific columns and it would be could to specify what the active cation binding phase is. Further discussion on the limitations of this binding phase for determining Ali would be useful, Would weakly complexed Al-organic species be retained on the column? This may need further discussion as if so it may mean less toxicity than what is assumed. This paper may have some useful information on this:

Robert W. Gensemer & Richard C. Playle (1999) The Bioavailability and Toxicity of

[Figure]

Aluminum in Aquatic Environments, Critical Reviews in Environmental Science and Technology, 29:4, 315-450, DOI: 10.1080/10643389991259245

Were any QA/QC checks performed on column performance (e.g. passing labile Al solution through and/or spikes).

Lines 163-167 - Also these speciation results appear quite similar to Simpson et al. (2014) from acid sulfate soil environments. Chemosphere 103, 172–180

Lines 189-194 - I suspect some of the lack of correlation is also due to a relatively low pH range within the data set. Hence any small uncertainties in analytical parameters such as pH become more influential. Maybe a little more caution could be applied to last sentence in this paragraph as this is not the case for some other streams globally. Before streamwaters I would suggest adding "acidified Nova Scotia..."

Some more discussion would be good also on the potential role of colloids affecting the results (and how they may bias measured Alo fraction?). I would suggest ultrafiltration could be used in future research to try to understand this further.

Table 1 - check subscripts on Al species

Figure 1 - suggest note in caption that shaded region corresponding to individual sites is the catchment area?

[Figure]

---

## Referee Comment (RC2) · Anonymous Referee #2 · 18 Jan 2020

General comments

Overall the paper presents an interesting dataset for a topic that probably hasn't received as much attention as it should have. The paper in its present form is largely site-specific case study, highlighting high concentrations of Al that exceed toxicity limits across several river basins, despite reductions in acid sources.

I think the paper can be suitable for publication, following some revision. The introduction sets a good background and context, but the methods and results could be mistaken for a monitoring report and could be enriched to better highlight the novelty in approach and the significance of the findings.

[Figure]

There are three main points I think could be considered to improve the paper: • I think the paper is missing a conceptual model. The mechanisms that could explain the observations are well described, but a conceptualisation of how they are inter-related could help frame the paper and give it a more general focus, making it more than just presentation of a site specific dataset – this is something that could be considered as a diagram for the discussion, and would help as a synthesis of the observations. • Related to the above, the paper could enrich the link to hydrology – seasonality is discussed at one point, but could this be expanded to better frame in context of catchment behaviour? I feel that after reading this paper, the take away message was that concentrations are high, above toxicity limits, which makes it perhaps more suited to an environmental quality journal, but what seems to be missing is how hydrology may be mediating concentrations and speciation. Again, this is where a conceptual model may help. • The statistics seems like it could benefit from a multi-variate approach to looking for patterns – currently the reliance on a series of independent correlations makes it hard to understand if there any interactions between variables. Use of GLMM is suggested at one point as a recommendation, but it's not clear why it wasn't done. For a paper in a top science journal, a full analysis of the data should be undertaken to come to a conclusion and show the results, rather than describe a possibility of showing something. In any event maybe there are other techniques that could be applied to better explain the variability?

Specific Comments

- Line 39-40. Sulfur emission reductions are mentioned here. But the paragraph opens straight away with acidification. Whilst most HESS readers I think will be familiar with the context, there may be readers without familiarity of this chemistry. I would therefore recommend extending the opening sentences to introduce the origin of acidification
- Following the above, the paper also maybe assumes the reader is familiar with the link between acidity and Al. Is this link predictable or depends on geology? Improving these contextual statement in the opening to highlight how sulfur, acid and aluminium

are related I think will help entrain readers (though I notice some coverage of this at the end of the intro). - Ln 46 – What is SWNS? - Ln 61 – Gibbsite is mentioned here. It would help to highlight the approximate pH where this kicks in (∼4.5?), and maybe replace "formation" with "precipitation" in case people are not aware it is a solid phase. - Ln 70 – This sentence seems like it should have come earlier (see initial comment): "Lowered pH increases Al solubility and observations confirm that Ali concentrations are negatively correlated with pH". Currently this paragraph on Al sources, comes after a paragraph on speciation and toxicity. - Line 96 – Should the ":" be a ";"? - Line 101 – I don't recall NS is defined by this point. - Line 110 – Here the aim of the paper is outlined, but it is a bit vague – there is a general desire to "increase understanding . . . " – could this be more specific? Ideally it would be good if the questions could also link to hydrology . . . eg is Al linked to hydrologic dynamics? - Line 147 – The description of statistics is brief. It looks like univariate statistics were done. It seems like the sort of dataset which requires a multivariate approach? PCA? - Line 149 – the last sentence introduces the term toxic threshold, but it doesn't follow from the previous sentences. Are you setting a threshold to determine exceedance frequency? Or are other metrics related to toxicity computed? - This toxicity value of 15 is mentioned, but I think it needs more justification and a clear rationale – is it acute or chronic, what is the origin and basis of this number? - Figure 1 & 2. I would have thought concentrations should come before the fraction % of samples above 15, as the latter has a higher level of interpretation. - Line 189 – Seems like a significant finding – could this be better highlighted as specific focus when framing your research question in the introduction, and also highlighting in the abstract? - Line 221 – it is suggested in the results that you should do a GLM model – why don't you do it in this study and present it here? I would have thought that for a hydrology focused journal, understanding the link between seasonality in hydrology and pH / Al relationship would be an important area to explore in detail, rather than just hint at it? - Conclusions – this is a good summary, but framed as 4 short paragraphs. I felt this could be more refined, maybe as just a single paragraph that flows better. Recommendations for further research on recovery

approach's may be useful?

---

## Author Comment (AC1) · 15 Feb 2020

We thank Anonymous Referee #1 for the thoughtful review of our manuscript. Below we respond (in bullets) to Anonymous Referee #1's comments (in quotes).

"Overall this is a well written manuscript that contributes new information on an important topic in relation to aquatic ecosystem risks from labile Al.

Line 134 - re: Bond Elut Jr column - I am unfamiliar with these specific columns and it would be could to specify what the active cation binding phase is. Further discussion on the limitations of this binding phase for determining Ali would be useful,
Would weakly complexed Al-organic species be retained on the column? This may need further discussion as if so it may mean less toxicity than what is assumed. This paper may have some useful information on this: Robert W. Gensemer & Richard C. Playle (1999) The Bioavailability and Toxicity of Aluminum in Aquatic Environments, Critical Reviews in Environmental Science and Technology, 29:4, 315-450, DOI: 10.1080/10643389991259245"

- We have added a discussion on the limitations of this binding phase for determining Ali, drawing upon Gensemer and Playle 1999. Weakly complexed Al-organic species, if positively charged, would be retained on the column (Aligent, personal communication). The assumption here is that if the cations are retained on the exchange column, they may also will be retained on the negatively charged fish gills, and therefore are potentially toxic.

"Were any QA/QC checks performed on column performance (e.g. passing labile Al solution through and/or spikes)."

- In addition to the standard lab QA/QC checks (described in Appendix D), we checked for column performance with spiked blank samples using ICP Al standard; the columns retained all of the aluminum. We have added the results of this QA/QC check to the description of the QA/QC checks.

"Lines 163-167 - Also these speciation results appear quite similar to Simpson et al. (2014) from acid sulfate soil environments. Chemosphere 103, 172–180."

- Thank you for bringing our attention to this paper. We have added this comparison to the results and use it to expand our discussion that, in addition to chronically acidified forests, acid sulphate soil environments also have elevated aluminum in drainage waters.

"Lines 189-194 - I suspect some of the lack of correlation is also due to a relatively low pH range within the data set. Hence any small uncertainties in analytical parameters

such as pH become more influential. Maybe a little more caution could be applied to last sentence in this paragraph as this is not the case for some other streams globally. Before streamwaters I would suggest adding "acidified Nova Scotia...""

- Good point about the lack of correlation between pH and Ali being likely due to a low pH range in observations. We have added this point to the discussion and have added more caution to the last sentence in the paragraph, defining more clearly the study area as chronically acidified. We have also highlighted other mechanisms that may cloud the strength of the inverse relationship between pH and Al: Al buffering in base cation-poor soils such as in Nova Scotia, increased DOC solubility at higher pH, increasing Al solubility in soils.

"Some more discussion would be good also on the potential role of colloids affecting the results (and how they may bias measured Alo fraction?). I would suggest ultrafiltration could be used in future research to try to understand this further."

- Simpson et al., 2014 is a useful example of how ultrafiltration can provide insights into colloidal forms of trace metals. To the discussion we have added a recommendation to use ultrafiltration and a description of how colloids might affect the results.

"Table 1 - check subscripts on Al species"

- Done and fixed.

"Figure 1 - suggest note in caption that shaded region corresponding to individual sites is the catchment area?"

- Done.

---

## Author Comment (AC2) · 15 Feb 2020

We thank Anonymous Referee #2 for the thoughtful review of our manuscript. Below we respond (in bullets) to Anonymous Referee #2's comments (in quotes).

"Overall the paper presents an interesting dataset for a topic that probably hasn't received as much attention as it should have. The paper in its present form is largely site-specific case study, highlighting high concentrations of Al that exceed toxicity limits across several river basins, despite reductions in acid sources.

I think the paper can be suitable for publication, following some revision. The introduction sets a good background and context, but the methods and results could be mistaken for a monitoring report and could be enriched to better highlight the novelty in approach and the significance of the findings.

There are three main points I think could be considered to improve the paper:

I think the paper is missing a conceptual model. The mechanisms that could explain the observations are well described, but a conceptualisation of how they are inter-related could help frame the paper and give it a more general focus, making it more than just presentation of a site specific dataset – this is something that could be considered as a diagram for the discussion, and would help as a synthesis of the observations."

- We agree. We have added a conceptual model. The paper is strengthened as a result. The conceptual model proposes new hypotheses to explain the insights into the drivers of toxic aluminium that we have learned from our study. Specifically, it proposes an explanation as to why, contrary to the existing conceptualizations of Ali concentrations being highest during spring peak flow, Ali concentrations are highest during summer low flow. And our Generalized Linear Mixed Model reveals, contrary to the standard conceptualization that DOC is inversely correlated with Ali, that DOC is strongly positively associated with Ali, suggesting that the increased recruitment of Al in soils by DOC may outweigh DOC's protective effects in waterbodies.

"Related to the above, the paper could enrich the link to hydrology – seasonality is discussed at one point, but could this be expanded to better frame in context of catchment behaviour? I feel that after reading this paper, the take away message was that concentrations are high, above toxicity limits, which makes it perhaps more suited to an environmental quality journal, but what seems to be missing is how hydrology may be mediating concentrations and speciation. Again, this is where a conceptual model may help."

- We agree. We have added a conceptual model and its discussion to the manuscript, illustrating how stormflow vs baseflow mediates concentrations and speciation.

"The statistics seems like it could benefit from a multi-variate approach to looking for patterns – currently the reliance on a series of independent correlations makes it hard to understand if there any interactions between variables. Use of GLMM is suggested at one point as a recommendation, but it's not clear why it wasn't done. For a paper in a top science journal, a full analysis of the data should be undertaken to come to a conclusion and show the results, rather than describe a possibility of showing something. In any event maybe there are other techniques that could be applied to better explain the variability?"

- Thank you for this insight. We have added a multivariate approach by developing a GLMM; the results support our conceptual model.

"Specific Comments

- Line 39-40. Sulfur emission reductions are mentioned here. But the paragraph opens straight away with acidification. Whilst most HESS readers I think will be familiar with the context, there may be readers without familiarity of this chemistry. I would therefore recommend extending the opening sentences to introduce the origin of acidification - Following the above, the paper also maybe assumes the reader is familiar with the link between acidity and Al. Is this link predictable or depends on geology? Improving these contextual statement in the opening to highlight how sulfur, acid and aluminium are related I think will help entrain readers (though I notice some coverage of this at the end of the intro)."

- We agree and have extended the opening sentences to introduce the origin of acidification, and have expanded the description of the link between acidity and Al.

"- Ln 46 – What is SWNS?"

- Thank you for catching this undefined acronym. It has been removed.

"- Ln 61 – Gibbsite is mentioned here. It would help to highlight the approximate pH where this kicks in (âĹij4.5?), and maybe replace "formation" with "precipitation" in case

people are not aware it is a solid phase."

- Thank you for catching this ambiguity. We have highlighted that gibbsite is formed at pH 6.0-8.0 at 25oC and approaching 6.5-8.5 at cooler temperatures (5oC) (Lydersen et al., 1990).

"- Ln 70 – This sentence seems like it should have come earlier (see initial comment): "Lowered pH increases Al solubility and observations confirm that Ali concentrations are negatively correlated with pH". Currently this paragraph on Al sources, comes after a paragraph on speciation and toxicity."

- Agreed, and fixed.

"- Line 96 – Should the ":" be a ";"?"

- Two independent clauses – yes, it should be a ";". Fixed.

"- Line 101 – I don't recall NS is defined by this point."

- We have added a definition of this acronym (Nova Scotia).

"- Line 110 – Here the aim of the paper is outlined, but it is a bit vague – there is a general desire to "increase understanding . . . " – could this be more specific? Ideally it would be good if the questions could also link to hydrology . . . eg is Al linked to hydrologic dynamics?"

- We have added more specific research questions linking hydrology to the observed chemical patterns.

"- Line 147 – The description of statistics is brief. It looks like univariate statistics were done. It seems like the sort of dataset which requires a multivariate approach? PCA?"

- Thank you for this insight. We have added a multivariate approach by developing a GLMM and have presented the results in the manuscript.

"- Line 149 – the last sentence introduces the term toxic threshold, but it doesn't follow

from the previous sentences. Are you setting a threshold to determine exceedance frequency? Or are other metrics related to toxicity computed? - This toxicity value of 15 is mentioned, but I think it needs more justification and a clear rationale – is it acute or chronic, what is the origin and basis of this number?"

- We have added a clarification of the origin and basis of the water quality threshold of 15 ug/L used in this manuscript; it is drawn from the FAO European Inland Fisheries Advisory Commission (EIFAC) water quality criteria for freshwater fish (Howells et al., 1990, Chemistry and Ecology). In the pH range of the 4.5-5.8, exposure of salmonids to concentrations of Ali around 10 to 15 $\mu$g L$-1$ for less than 14 days were found to acutely impair the sea-water tolerance of salmon smolts (Staurnes et al., 1995; Howells et al., 1990; Dennis and Clair, 2012; Kroglund et al., 2012).

"- Figure 1 & 2. I would have thought concentrations should come before the fraction % of samples above 15, as the latter has a higher level of interpretation."

- We agree that current Figure 2 should go before Figure 1; further, to reduce redundancies, we have combined Figures 1 and 2 into one figure with two panels.

"- Line 189 – Seems like a significant finding – could this be better highlighted as specific focus when framing your research question in the introduction, and also highlighting in the abstract?"

- Reviewer #1 pointed out that the lack of correlation between pH and Ali is likely due to a low pH range in observations. We have added this point to the discussion. We have also highlighted other mechanisms that may cloud the strength of the inverse relationship between pH and Al: Al buffering in base cation-poor soils such as in Nova Scotia, increased DOC solubility at higher pH, increasing Al solubility in soils. In contrast, our finding that DOC is significantly and positively correlated with Ali, rather than inversely - as has been stated in the literature (e.g., Gensenmer and Playle, 1999), is important. This finding has important policy implications. For decades, policy makers in Nova Scotia assumed that the high levels of DOC in Nova Scotian rivers protected

wild Atlantic salmon populations from aluminum. We have improved the highlighting of this significant finding in the conceptual model, research question and in the abstract.

"- Line 221 – it is suggested in the results that you should do a GLM model – why don't you do it in this study and present it here? I would have thought that for a hydrology focused journal, understanding the link between seasonality in hydrology and pH / Al relationship would be an important area to explore in detail, rather than just hint at it?"

- We agree. We have developed a GLMM to explore these ideas in more detail and have presented the results in the revised manuscript. The model results reinforce that DOC concentrations are positively associated with cationic aluminum on a seasonal basis.

"- Conclusions – this is a good summary, but framed as 4 short paragraphs. I felt this could be more refined, maybe as just a single paragraph that flows better. Recommendations for further research on recovery approach's may be useful?"

- We agree. We have changed the conclusion to a single paragraph and have added recommendations for further research on recovery approaches.

---

## Author Response (AR1)

[revised manuscript text omitted]

Table A3

**Table A3** Generalized linear mixed model (GLMM) results for complete field data.

| Fixed Effect | Parameter Estimate | Wald t Test Statistic | P-Value | AIC |
|---|---|---|---|---|
| Ca | 0.281 | 1.551 | 0.121 | |
| DOC | 0.536 | 3.285 | **0.001** | |
| F | -0.04 | -0.79 | 0.429 | |
| NO3 | 0.068 | 3.269 | **0.001** | 1316.9 |
| pH | -1.123 | -0.952 | 0.341 | |
| SO4 | -0.295 | -3.038 | **0.002** | |
| Tw | 0.34 | 1.551 | **0.046** | |
| DOC | 0.321 | 5.647 | **0** | 1946.3 |
| DOC | 0.149 | 4.954 | **0** | |
| NO3 | 0.417 | 2.721 | **0.007** | 1816.7 |
| SO4 | -0.417 | -2.667 | **0.008** | |
| DOC | 0.256 | 6.908 | **0** | |
| NO3 | 0.12 | 3.335 | **0** | 1837.2 |
| DOC*NO3 | 1.1 | 4.545 | **0** | |
| Tw | 0.548 | 4.574 | **0** | 1467.8 |
| DOC | 1.135 | 3.445 | **0** | |
| Tw | 0.678 | 2.215 | **0.027** | 1438.2 |
| DOC*Tw | -0.470 | 0.109 | 0.109 | |
| DOC | 0.623 | 6.391 | **0** | 1438.6 |
| Tw | 0.24 | 1.943 | *0.052* | |

-significant parameters at the 5% significance level are bolded

-significant parameters at the 10% significance level are italicized

-Effect connected by "*" represent an interaction term.

Table A4 Linear correlation $r^2$ values and significance ($\alpha = 0.05$) between $Al_i/Al_d$ and other water chemistry parameters across all sites.

| Variable | Unit | Correlation with $Al_i/Al_d$ ($R^2$) | Significance (p-value) |
|---|---|---|---|
| $Al_d$ | $\mu g$ $L^{-1}$ | 0.007 | 0.247 |
| Ca | $\mu g$ $L^{-1}$ | 0.001 | 0.676 |
| DOC | mg $L^{-1}$ | 0.007 | 0.247 |
| pH | unit | 0.077 | 0.000 |
| Water Temp. | °C | 0.114 | 0.000 |
| $F^+$ | $\mu g$ $L^{-1}$ | 0.003 | 0.537 |
| $NO_3^-$ | $\mu g$ $L^{-1}$ | 0.002 | 0.624 |
| $SO_4^{2-}$ | $\mu g$ $L^{-1}$ | 0.000 | 0.952 |

Table A4A5  Kendal-tau correlation and significance (α = 0.05) between $Al_i$ and other water chemistry parameters for each study site.  One $Al_i$ outlier removed for MR calculations (value: 2 µg $L^{-1}$, date: 30 April 2015).

| Site | Variable | Unit | Slope | Correlation | Significance (p-value) |
|---|---|---|---|---|---|
| ALD | Ald | µg $L^{-1}$ | | 0.29 | 0.044 |
| | Ca | µg $L^{-1}$ | | 0.22 | 0.143 |
| | DOC | mg $L^{-1}$ | | 0.36 | 0.013 |
| | pH | unit | | 0.19 | 0.190 |
| | Water Temp. | °C | | 0.32 | 0.093 |
| | $F^+$ | µg $L^{-1}$ | | 0.182 | 0.533 |
| | $NO_3^-$ | µg $L^{-1}$ | | 0.600 | 0.142 |
| | $SO_4^{2-}$ | µg $L^{-1}$ | | -0.037 | 0.876 |
| BLB | Ald | µg $L^{-1}$ | | 0.03 | 0.852 |
| | Ca | µg $L^{-1}$ | | 0.17 | 0.238 |
| | DOC | mg $L^{-1}$ | | 0.08 | 0.575 |
| | pH | unit | | 0.07 | 0.622 |
| | Water Temp. | °C | | 0.35 | 0.099 |
| | $F^+$ | µg $L^{-1}$ | | -0.036 | 0.901 |
| | $NO_3^-$ | µg $L^{-1}$ | | -0.109 | 0.708 |
| | $SO_4^{2-}$ | µg $L^{-1}$ | | -0.184 | 0.468 |
| CC | Ald | µg $L^{-1}$ | | 0.11 | 0.708 |

| | | | | | |
|---|---|---|---|---|---|
| | | | µg | | |
| | Ca | L⁻¹ | | -0.22 | 0.451 |
| | | | mg | | |
| | DOC | L⁻¹ | | 0.25 | 0.383 |
| | pH | | unit | -0.04 | 0.901 |
| | Water | | | | |
| | Temp. | | °C | 0.67 | 0.174 |
| | F+ | µg | | | |
| | | L⁻¹ | | | |
| | NO₃⁻ | µg | | | |
| | | L⁻¹ | | | |
| | SO₄²⁻ | µg | | | |
| | | L⁻¹ | | | |
| | | | µg | | |
| | Ald | L⁻¹ | | 0.800 | 0.050 |
| | | | µg | | |
| | Ca | L⁻¹ | | 0.200 | 0.624 |
| | | | mg | | |
| | DOC | L⁻¹ | | 0.800 | 0.050 |
| | pH | | unit | -0.200 | 0.624 |
| KB | Water | | | | |
| | Temp. | | °C | 0.600 | 0.142 |
| | F+ | µg | | 0.800 | 0.050 |
| | | L⁻¹ | | | |
| | NO₃⁻ | µg | | | |
| | | L⁻¹ | | | |
| | SO₄²⁻ | µg | | -0.400 | 0.327 |
| | | L⁻¹ | | | |
| | | | µg | | |
| | Ald | L⁻¹ | | 0.37 | 0.047 |
| | | | µg | | |
| | Ca | L⁻¹ | | 0.24 | 0.226 |
| LR | | | mg | | |
| | DOC | L⁻¹ | | 0.25 | 0.189 |
| | pH | | unit | 0.19 | 0.319 |
| | Water | | | | |
| | Temp. | | °C | 0.02 | 0.937 |

| Group | Parameter | Unit | | |
|---|---|---|---|---|
| | F+ | $\mu g\ L^{-1}$ | | |
| | $NO_3^-$ | $\mu g\ L^{-1}$ | -0.333 | 0.348 |
| | $SO_4^{2-}$ | $\mu g\ L^{-1}$ | 0.105 | 0.801 |
| MB | Ald | $\mu g\ L^{-1}$ | 0.739 | 0.001 |
| | Ca | $\mu g\ L^{-1}$ | -0.062 | 0.783 |
| | DOC | $mg\ L^{-1}$ | 0.400 | 0.073 |
| | pH | unit | -0.279 | 0.214 |
| | Water Temp. | °C | 0.125 | 0.580 |
| | F+ | $\mu g\ L^{-1}$ | -0.028 | 0.917 |
| | $NO_3^-$ | $\mu g\ L^{-1}$ | -0.182 | 0.533 |
| | $SO_4^{2-}$ | $\mu g\ L^{-1}$ | -0.463 | 0.050 |
| MPB | Ald | $\mu g\ L^{-1}$ | 0.550 | 0.000 |
| | Ca | $\mu g\ L^{-1}$ | 0.580 | 0.000 |
| | DOC | $mg\ L^{-1}$ | 0.574 | 0.000 |
| | pH | unit | -0.169 | 0.146 |
| | Water Temp. | °C | 0.280 | 0.016 |
| | Runoff | $mm\ day^{-1}$ | -0.232 | 0.042 |
| | F+ | $\mu g\ L^{-1}$ | 0.239 | 0.042 |
| | $NO_3^-$ | $\mu g\ L^{-1}$ | 0.190 | 0.160 |

| | | | | | |
|---|---|---|---|---|---|
| | SO$_4$$^{2-}$ | L$^{-1}$ | µg | -0.206 | 0.067 |
| MR | Ald | L$^{-1}$ | µg | 0.459 | 0.000 |
| | Ca | L$^{-1}$ | µg | 0.317 | 0.002 |
| | DOC | L$^{-1}$ | mg | 0.382 | 0.000 |
| | pH | | unit | 0.097 | 0.362 |
| Temp. | Water Temp. | | °C | 0.285 | 0.007 |
| | RunOff | day$^{-1}$ | mm | -0.108 | 0.291 |
| | F+ | L$^{-1}$ | µg | 0.139 | 0.188 |
| | NO$_3$$^-$ | L$^{-1}$ | µg | 0.086 | 0.450 |
| | SO$_4$$^{2-}$ | L$^{-1}$ | µg | -0.127 | 0.215 |
| PMB | Ald | L$^{-1}$ | µg | 0.46 | 0.019 |
| | Ca | L$^{-1}$ | µg | 0.01 | 0.960 |
| | DOC | L$^{-1}$ | mg | 0.21 | 0.295 |
| | pH | | unit | -0.23 | 0.232 |
| Temp. | Water Temp. | | °C | 0.36 | 0.065 |
| | F+ | L$^{-1}$ | µg | -0.063 | 0.782 |
| | NO$_3$$^-$ | L$^{-1}$ | µg | 0.276 | 0.444 |
| | SO$_4$$^{2-}$ | L$^{-1}$ | µg | -0.293 | 0.135 |
| UKR | Ald | L$^{-1}$ | µg | 0.34 | 0.071 |

| | | | | |
|---|---|---|---|---|
| | Ca | µg L$^{-1}$ | 0.38 | 0.053 |
| | DOC | mg L$^{-1}$ | 0.32 | 0.086 |
| | pH | unit | 0.35 | 0.063 |
| Temp. | Water | °C | 0.14 | 0.621 |
| | F+ | µg L$^{-1}$ | | |
| | NO$_3^-$ | µg L$^{-1}$ | | |
| | SO$_4^{2-}$ | µg L$^{-1}$ | -0.600 | 0.142 |

**Table** **A6** Generalized linear mixed model (GLMM) results for seasonal field data.

| Fixed Effect | Parameter Estimate | Wald t Test Statistic | P-Value | AIC |
|---|---|---|---|---|
| Ald | 0.264 | 6.17 | **0** | |
| Ca | -0.007 | -0.183 | 0.855 | |
| DOC | 0.143 | 3.727 | **0** | 1736.5 |
| F | -0.020 | -0.207 | 0.836 | |
| NO3 | 0.146 | 0.991 | 0.322 | |
| SO4 | -0.133 | -1.129 | 0.259 | |
| ALd | 0.281 | 6.921 | **0** | 1867.3 |
| DOC | 0.078 | 1.877 | *0.061* | |
| ALd | 0.313 | 7.393 | **0** | |
| DOC | 0.158 | 3.152 | **0.002** | 1862.8 |
| ALd*DOC | -0.076 | -2.490 | **0.013** | |
| ALd | 0.332 | 11.49 | **0** | 1868.3 |
| DOC | 0.229 | 9.445 | **0** | 1909.9 |
| DOC | 0.247 | 9.744 | **0** | |
| NO3 | 0.329 | -2.399 | **0.016** | 1768.4 |
| SO4 | -0.316 | 2.515 | **0.012** | |
| DOC | 0.287 | 9.453 | **0** | |
| NO3 | 0.063 | 1.733 | *0.083* | 1797.3 |
| DOC*NO3 | 0.41 | 1.709 | *0.088* | |

-significant parameters at the 5% significance level are bolded

-significant parameters at the 10% significance level are italicized

-Effect connected by "*" represent an interaction term.

[revised manuscript text omitted]

**Script for GLMM model**

```
**setwd**
setwd("C:\\Users\\50nlo\\Documents\\Research\\MS_AliPatterns\\Dat
a")

**load packages**
**require(lme4)**
require(car)
require(MASS)

**Read in Data**
ALiDatDF<-as.data.frame(read.csv("GLMM_Input_V2.csv",header=T))
ALiDat<-ALiDatDF$Ali_ugL

**Exploratory data analysis of ALi data**
hist(ALiDat) #data are skewed
```

```
**Test goodness-of-fit of lognormal data**

**Normal QQ plot for comparison**

qqnorm(ALiDat)

qqline(ALiDat)

qqp(ALiDat, "norm")

**lognormal QQ plot**

fit_params <- fitdistr(ALiDat,"lognormal")

quants <-seq(0,1,length=length(ALiDat))[2:138]

fit_quants <- qlnorm(quants,fit_params$estimate['meanlog'],
fit_params$estimate['sdlog'])

data_quants <- quantile(ALiDat,quants)

plot(fit_quants, data_quants, xlab="Theoretical Quantiles",
ylab="Sample Quantiles")

title(main = "Q-Q plot of lognormal fit against data")

abline(0,1)

qqp(ALiDat, "lnorm")

**Gamma QQ plot**
```

```
gamma <- fitdistr(ALiDat, "gamma")

qqp(ALiDat, "gamma", shape = gamma$estimate[[1]], rate =
gamma$estimate[[2]])

**Exponential QQ plot**

exp <- fitdistr(ALiDat, "exponential")

qqp(ALiDat, "exp", rate = gamma$estimate[[1]])

Site<-ALiDatDF$Site

Season<-ALiDatDF$Season

ALd<-scale(ALiDatDF$Ald_ugL)

Ca<-scale(ALiDatDF$Ca_ugL)

DOC<-scale(ALiDatDF$DOC_mgL)

pH<-scale(ALiDatDF$Calib_pH)

SO4<-scale(ALiDatDF$SO4_ugL)

Tw<-scale(ALiDatDF$Tw_C)

F<-scale(ALiDatDF$F_ugL)

NO3<-scale(ALiDatDF$NO3_ugL)

Dis<-scale(ALiDatDF$Disch_m3s)

**ALd and Season cause a singular fit (overfit)**

**This means that the effect structure is too complex to be**
supported by the data

**ALd is likely due to it being a function of ALi**
```

```
**Season is due to limited seasonal data at each site.**

Models <- glmer(ALiDat ~ DOC + Tw + Ca+ pH + SO4 + F + NO3 + (1 |
Site), family = gaussian(link = "log"), control=glmerControl(optimizer="bobyqa",optCtrl=list(maxfun=2e5)))

    summary(Models)

    Models <- glmer(ALiDat ~ DOC + SO4 + NO3 + (1 | Site), family =
gaussian(link = "log"), control=glmerControl(optimizer="bobyqa",optCtrl=list(maxfun=2e5)))

    summary(Models)

    Models <- glmer(ALiDat ~ DOC + NO3 + DOC*NO3+(1 | Site), family =
gaussian(link = "log"), control=glmerControl(optimizer="bobyqa",optCtrl=list(maxfun=2e5)))

    summary(Models)

    Models <- glmer(ALiDat ~ DOC +(1 | Site), family = gaussian(link
= "log"), control=glmerControl(optimizer="bobyqa",optCtrl=list(maxfun=2e5)))

    summary(Models)

    Models <- glmer(ALiDat ~ Tw +(1 | Site), family = gaussian(link =
"log"))

    summary(Models)
```

```
Models <- glmer(ALiDat ~ DOC + Tw +(1 | Site), family =
gaussian(link = "log"))
summary(Models)

Models <- glmer(ALiDat ~ DOC + Tw + DOC*Tw + (1 | Site), family =
gaussian(link = "log"))
summary(Models)

**pH and Tw causes a singular fit (overfit)**
Models <- glmer(ALiDat ~ ALd + DOC + Ca + SO4 + F + NO3 + (1 |
Season), family = gaussian(link = "log"))
summary(Models)

Models <- glmer(ALiDat ~ ALd + DOC + (1 | Season), family =
gaussian(link = "log"))
summary(Models)

Models <- glmer(ALiDat ~ ALd + DOC + ALd*DOC + (1 | Season),
family = gaussian(link = "log"))
summary(Models)

Models <- glmer(ALiDat ~ ALd + (1 | Season), family =
gaussian(link = "log"))
```

```
summary(Models)

Models <- glmer(ALiDat ~ DOC + (1 | Season), family =
gaussian(link = "log"))
summary(Models)

Models <- glmer(ALiDat ~ DOC + SO4 + NO3 + (1 | Season), family =
gaussian(link = "log"))
summary(Models)

Models <- glmer(ALiDat ~ DOC + NO3 + DOC*NO3+(1 | Season), family
= gaussian(link = "log"))
summary(Models)

################################For
testing############################
**95% confidence intervals**
fm1W <- confint.merMod(Models, method="Wald")

**Check for singularity**
tt <- getME(Models,"theta")
ll <- getME(Models,"lower")
min(tt[ll==0])

**Use penalized quazilikelihood to estimate non-normal parameters**
```

```
PQL <- glmmPQL(ALiDat ~ ALd + DOC + pH + SO4+ Tw + NO3 + Ca, ~1 |
Site, family = gaussian(link = "log"), verbose = FALSE)

**Fluoride is confounded, remove from model.**

summary(PQL)

**At the 5% sig. level, pH, SO4, NO3, and Ca are not significant**
effects

**Use penalized quazilikelihood to estimate non-normal parameters**
PQL <- glmmPQL(ALiDat ~ ALd + Tw + DOC + ALd*pH + Tw*pH, ~1 |
Site, family = gaussian(link = "log"), verbose = FALSE)
summary(PQL)

resid<-as.matrix(PQL$residuals[,1])
**Explore the model residuals**
acf(resid) #good
```

**Appendix D. Additional methods**

**D.1 Laboratory analysis methods**

Samples were analyzed at Maxxam Analytics Laboratory, Health and Environmental Research Centre (HERC), and AGAT Laboratories. Samples from MR, MPB, PMB, MB, KB, and CC were analyzed at Maxxam and HERC labs only. Samples from BLB, ALD, UKR, and LR were analyzed at all three labs.

**D1.1 Maxxam Laboratory**

The protocol at Maxxam Laboratory in Bedford, NS, adheres to methods approved by the United States Environmental Protection Agency (US EPA) for identifying trace elements in water (US EPA, 1994) and analyzing samples using Inductively Coupled Plasma-Mass Spectrometry (ICP-MS) (US EPA, 1998). Cations and anions were analyzed using ICP-MS, while a Continuous Flow Analyzer was used to measure DOC. pH was measured using a standard hydrogen electrode and reference electrode.

**D1.2 HERC Laboratory**

$SO_4^{2-}$ samples were analyzed at HERC Laboratory in Halifax, NS, due to lower detection limits at the Maxxam laboratory. Once delivered to the laboratory, samples were filtered using a 0.45 µm glass fiber filter and analyzed using an Ion-Chromatography System (ICS) 5000 Dionex detector.

**D1.3 AGAT Laboratory**

Samples collected in the West River, Sheet Harbour area (UKR, ALD, LR, BLB, KB, CC) were analyzed at the AGAT laboratory in Dartmouth, NS. This laboratory holds the

9001:2015 and 17025:2005 International Organization for Standardization accreditations. Cation samples were analyzed using ICP-MS, laboratory pH was measured using a standard hydrogen electrode and reference electrode, and $SO_4^{2-}$ and anions were measured using ICS. Samples analyzed at AGAT were analyzed for total organic carbon (TOC) as opposed to DOC and were analyzed using Infrared Combustion (IR Combustion).

**D.2 Data quality assurance and control**

Blanks were used to assess contamination during the $Al_o$ extraction procedure. Blanks were collected on 10% of samples, taken on arbitrary sampling events. Triple deionized water was collected before passing through filter and column ("Blank Before"), and after ("Blank After"). The triple-deionized water had traces of chemicals below the laboratory detection limits, providing "Not Detectable" results for the Blank Before sample. If chemicals were detected in the Blank After sample, this would have indicated leaching of chemicals from the column.

Duplicates were collected and analyzed for 10% of the samples; on arbitrarily selected sampling events, $Al_o$ and $Al_{filtered}$ or $Al_{unfiltered}$, were analyzed twice, independently, by Maxxam laboratory. All laboratories also conducted additional duplicate, blank, reference material, and matrix spike testing, in addition to instrument calibration in adherence to industry standards for quality control and assurance.

Spiked blank samples were conducted using ICP Al standard, 1000 ug/mL, $HNO_3$ (SCP Science). Three types of measurements were taken. The 'total' measurement was an unaltered sample of the diluted solution created above. The 'dissolved' measurement was a sample of the above solution passed through a 0.45um PES filter. The 'organic' measurement as a sample of the above solution passed through a 0.45um PES filter and a cation exchange column.

The spiked column blanks show that the columns are performing well; the cation exchange column removed virtually all of the Al in the solution (detection limit = 4 ug/L). Additional blanks were conducted in the Dalhousie hydrology lab that was used to prepare the sampling equipment before field collection.  The blanks showed no contamination.

To verify that sample analysis results from the Maxxam/HERC laboratory combination were comparable to AGAT, three sets of duplicate samples were collected for ALD, BLB, UKR, and LR (19 April 2017, 14 May 2017, and 30 May 2017) and analyzed by both laboratories. Laboratory results were compared using Wilcoxon Rank Sum statistical test in Python 3.6.5 using the SciPy Stats module (version 0.19) (Appendix C.2). Results indicated a significant difference in pH values between laboratories ($T = 1$, $p = 0.04$), therefore, statistical analysis on pH data was conducted on the calibrated YSI Pro Plus sonde field data. $Al_o$, $Al_{filtered}$, and $Al_{unfiltered}$ results were found to be comparable between laboratories ($T = 8.5$, $p = 0.674$; $T = 5.0$, $p = 0.249$; and $T = 8.0$, $p = 0.600$, respectively). After adjusting for detection limits (Table A6), Ca results were also found to be comparable between laboratories ($T = 4.0$, $p = 0.173$). However, due to the large difference in $SO_4^{2-}$ detection limits between HERC and AGAT (10 µg $L^{-1}$ and 2 mg $L^{-1}$, respectively), results for $SO_4^{2-}$ are not comparable between laboratories. Lastly, organic carbon analyzed at Maxxam was analyzed for DOC, while AGAT analyzed for TOC, therefore these results cannot be compared. For dates where duplicate data is present, AGAT data was used to maintain data source consistency, apart from $SO_4^{2-}$ data, for which HERC data was used due to superior detection limits. Analysis for BLB and ALD transitioned from Maxxam to AGAT 19 April 2017 and consequently DOC is approximated as TOC for these two sites after this date.

The YSI Pro Plus sonde was calibrated within 36 hours of in-stream data collection.

**D.3 Toxic thresholds of $Al_i$**

Identified toxic thresholds of $Al_i$ for *Salmo salar* vary in the literature. Based on toxicological and geochemical studies on Al and *Salmo salar*, the EIFAC suggested an $Al_i$ toxic threshold of 15 ug $L^{-1}$ for Atlantic salmon in freshwaters for pH between 5.0 and 6.0, and 30 ug $L^{-1}$ in pH <5 (Howells et al., 1990). The lower threshold at higher pH is to account for the increased fraction in the $Al(OH)_2^+$ species. At pH > 6, the toxic effects of $Al_i$ to *Salmo salar* are considered negligible, and toxic effects are dominated by other dissolved and precipitated forms (Gensemer et al., 2018), due to the decreased solubility of Al at pH > 6 (Dennis and Clair 2012). However, in colder rivers, the pH-toxicity threshold may be higher, closer to pH 6.5 (Lydersen, 1990). For the purposes of this study, we use the toxic threshold of $Al_i$ at 15 ug $L^{-1}$, as the majority of our pH observations were greater than or equal to 5.0 (Table A2).

**D.4 Calibration of pH measurements**

In situ pH measurements were taken using a YSI Pro Plus sonde and confirmed with a YSI Ecosense pH Pen. It was found that measurements taken with the YSI Pro Plus sonde deviated from the YSI Ecosense Pen, which is known to measure pH accurately ($0.47 \pm 0.44$ pH units below in-stream pH as measured by YSI Ecosense Pen). Therefore, a calibration curve was created based on simultaneous side-by-side measurements of both instruments (n = 69 pairs) and the in situ pH data were adjusted accordingly (Eq. 1).

$$YSI\ Ecosense\ Pen\ pH = 0.595(Pro\ Plus\ pH) + 2.3868 \qquad (1)$$